# Output Feedback Integrated Guidance and Control Design for Autonomous Underwater Vehicles Against Maneuvering Targets

**DOI:** 10.3390/s25103088

**Published:** 2025-05-13

**Authors:** Rui Wang, Jingwei Lu, Shuke Lyu, Yongtao Liu, Yuchen Cui

**Affiliations:** 1School of Emergency Equipment, North China Institute of Science and Technology, Beijing 101601, China; ruiwang@ncist.edu.cn (R.W.); sk_lv@foxmail.com (S.L.); cuiyuchen2003@ncist.edu.cn (Y.C.); 2School of Information and Control Engineering, North China Institute of Science and Technology, Beijing 101601, China; ytliu@ncist.edu.cn; 3Key Laboratory of Special Robots for Safety Production and Emergency Disposal in Hebei Province, North China Institute of Science and Technology, Langfang 065201, China; 4Key Laboratory of Safety Monitoring of Mining Equipment in Hebei Province, North China Institute of Science and Technology, Langfang 065201, China; 5Department of Industrial Engineering, Tsinghua University, Beijing 100084, China

**Keywords:** integrated guidance and control, autonomous underwater vehicles, finite-time extended state observer, sliding mode control, event-triggered control

## Abstract

Traditional guidance and control systems often treat guidance and control systems separately, leading to reduced interception accuracy and responsiveness, especially during high-speed terminal trajectories. These limitations are further exacerbated in autonomous underwater vehicles (AUVs) due to unknown wave/current disturbances, harsh underwater acoustic conditions, and limited sensor capabilities. To address these challenges, this paper studies an integrated guidance and control (IGC) design for AUVs intercepting maneuvering targets with unknown disturbances and unmeasurable system states. The IGC model is derived based on the relative motion equations between the AUV and the target, incorporating the lateral dynamics of the AUV. A model transformation is introduced to synthesize external disturbances with unmeasurable states, extending the resultant disturbance to a new system state. A finite-time convergent extended state observer (ESO) is thus designed for the transformed system to estimate the unknown signals. Using these estimates from the observer, a finite-time event-triggered sliding mode controller is developed, ensuring finite-time convergence of system errors to an adjustable residual set, as rigorously proven through Lyapunov stability analysis. Simulation results demonstrate the superiority of the proposed method in achieving higher interception accuracy and faster response compared to traditional guidance and control approaches with unknown disturbances and unmeasurable states.

## 1. Introduction

In the field of ocean engineering, high-speed AUVs play an important role in intercepting underwater targets such as anti-ship AUVs or missiles [1,2,3,4]. A key factor influencing interception accuracy is the guidance and control system. Traditional approaches treated the guidance control systems as separate entities, where the guidance system computed required guidance commands based on relative motion dynamics, by which the control system generated the actual control rudder inputs to execute the interception. While these methods have been extensively studied [5,6,7,8,9,10,11], they overlooked the mutual coupling and hysteresis between the guidance and control systems. This limitation becomes particularly problematic during high-speed terminal trajectories, where rapid attitude changes lead to larger miss distances and reduced hitting accuracy.

IGC addresses these challenges by unifying the design of guidance and control systems. By directly generating control commands based on relative motion dynamics and dynamic model characteristics [12,13,14], IGC enhances both interception accuracy and responsiveness. Initially developed for missile flight control systems [15], IGC has since been extended to address nonlinear disturbances [16], modeling errors [17], and composite disturbances [18]. Notable advancements include finite-time estimation of disturbances using second-order sliding mode methods [18], non-singular terminal sliding mode control [19], parallel control [20,21], and bounded control strategies to prevent actuator saturation [22].

Despite these advances, most IGC schemes are tailored for missiles and aircraft, which are not directly applicable to AUVs operating in complex marine environments. Challenges specific to AUVs include unknown wave/current disturbances, harsh underwater acoustic conditions, and limited sensor capabilities [1,23]. For instance, AUVs cannot directly measure sideslip angle or line-of-sight (LOS) velocity due to the fixed acoustic transducer array firmly attached to the shell, unlike missiles and aircrafts that rely on infrared and radio sensors. Additionally, the short interception window and high-speed maneuverability of underwater targets exacerbate the need for fast response systems. Failure to account for guidance-control hysteresis can lead to significant miss distances [24,25,26,27], underscoring the necessity of IGC designs for AUVs.

The motivation of this study stems from the critical need to enhance the interception accuracy and responsiveness of AUVs in complex marine environments. Existing solutions often involve designing observers to estimate unmeasurable signals [28,29,30,31], but these methods typically ignore the impact of external disturbances on estimation accuracy. Furthermore, Extended state observers (ESOs) have demonstrated robustness in estimating unknown disturbances and target maneuvers [32,33,34,35], which lumped the unknown signals together as new states that can be estimated online, but these methods developed for aircraft guidance and control systems cannot be directly applied to AUVs. These gaps in the literature motivate the development of a novel IGC framework for AUVs.

Motivated by the aforementioned challenges, this paper proposes an IGC design for AUVs intercepting maneuvering targets with unknown disturbances and unmeasurable system states. A finite-time ESO is constructed to estimate the unmeasured signals, including disturbances, sideslip angle, LOS velocity, and target maneuvers, by which an event-triggered sliding mode controller is then designed to realize a successful interception. Meanwhile, event-triggered control strategies [36,37] reduce unnecessary actuator activations while preserving system stability and performance, which aligns with the energy-efficiency requirements of the AUVs. The contributions of this paper are mainly threefold:State Transformation: Accounting for external disturbances and unmeasurable states, a novel state transformation is proposed to convert the AUV IGC model with unmatched disturbances into a system with matched disturbances, thereby facilitating effective observer and controller design.Finite-Time ESO Design: Based on the transformed system, a finite-time ESO is developed to estimate the unknown states and disturbances in finite-time, ensuring robustness against external perturbations.Event-Triggered IGC Controller: Using the ESO estimates, the IGC design is proposed such that a successful interception of the incoming target is realized with unknown disturbances and the unmeasurable sideslip angle and LOS velocity. The event-triggering mechanism reduces communication and actuator burdens, enhancing system efficiency.

The proposed method demonstrates superior performance in simulation studies, achieving higher interception accuracy and faster response compared to traditional guidance and control approaches with unknown disturbances and unmeasurable states.

The remainder of this paper is organized as follows. Section 2 formulates the IGC design problem and provides necessary preliminaries. Section 3 details the state transformation, finite-time ESO design, and controller development. Simulation results and discussions are presented in Section 4, followed by conclusions in Section 5.

## 2. Preliminaries

### Problem Formulation

Notice that the depth range of the target varies very little in the terminal phase; it is assumed that the AUV and the target are moving in the same plane. The interception geometry between the AUV and the target is illustrated in Figure 1, where A and $T_{g}$ denote the positions of the AUV and target, and the axes of *x* and *y* are north and east directions of the Earth-fixed frame, respectively. The reference line $x_{0}$ is parallel to the *x* axis with the same direction. Additionally, *q* is the LOS angle, $\sigma_{A}$ is the ballistic inclination angle, $\sigma_{T}$ is the target course angle, $\eta_{A}$ is the advanced angle of the AUV velocity, and $\eta_{T}$ is the advanced angle of the target velocity. The starting edge of all the angles is $x_{0}$, such that the positive direction of the angle is anti-clockwise from $x_{0}$ to the adjacent side; *r* is the relative distance between the AUV and the target; and $r_{min}>0$ is the maximum allowable miss, i.e., successful target interception is achieved when $|r|<r_{min}$.

According to the relative motion relationship between the AUV and the target shown in Figure 1, the AUV velocity $v_{A}$ and the target velocity $v_{T}$ are decomposed along the LOS direction and its normal direction, respectively, such that the relative motion equations are given by(1)$$\begin{array}{cc} \; & \dot{r}=-v_{A}\cos\left(q-\sigma_{A}\right)+v_{T}\cos\left(q-\sigma_{T}\right)\\ \; & \dot{q}=\frac{1}{r}\left(v_{A}\sin\left(q-\sigma_{A}\right)-v_{T}\sin\left(q-\sigma_{T}\right)\right) \end{array}$$

Notice that the AUV velocity and the target velocity are almost constants in the end trajectory. Thus, we obtain(2)$${\dot{v}}_{A}=0,\;{\dot{v}}_{T}=0$$

Taking the time-derivative of (Equation 1) yields(3)$$\begin{array}{cc} \; & \ddot{r}=-v_{A}{\dot{\sigma }}_{A}\sin\left(q-\sigma_{A}\right)+v_{T}{\dot{\sigma }}_{T}\sin\left(q-\sigma_{T}\right)+r{\dot{q}}^{2}\\ \; & r\ddot{q}=-2\dot{r}\dot{q}-v_{A}{\dot{\sigma }}_{A}\cos\left(q-\sigma_{A}\right)+v_{T}{\dot{\sigma }}_{T}\cos\left(q-\sigma_{T}\right) \end{array}$$

The disturbance term $\Delta_{q}$ caused by the target manuevering is defined as(4)$$\Delta_{q}≜v_{A}{\dot{\sigma }}_{A}\left(1-\cos\left(q-\sigma_{A}\right)\right)+v_{T}{\dot{\sigma }}_{T}\cos\left(q-\sigma_{T}\right)$$
where the first term denotes the measurement error of *q* and $\sigma_{A}$, and the second term is the unmeasurable target motion information.

Using (Equation 4), the second equation of (Equation 3) equals to(5)$$r\ddot{q}=-2\dot{r}\dot{q}-v_{A}{\dot{\sigma }}_{A}+\Delta_{q}$$

According to the results in [38,39,40], the dynamic model equations of the AUV lateral channel are given by(6)$$\begin{array}{cc} {\dot{\sigma }}_{A}= & \frac{1}{A_{x}v_{A}}\left(-K_{z\beta }v_{A}^{2}+\frac{\lambda_{35}K_{my\beta }v_{A}^{2}}{K_{J}}+T\right)\cdot \beta \\ \; & \;+\frac{1}{A_{x}}\left(-K_{z\omega }+\lambda_{33}+\frac{\lambda_{35}K_{my\omega }-\lambda_{35}^{2}}{K_{J}}\right)\cdot \omega_{y}\\ \; & \;+\frac{v_{A}}{A_{x}}\left(-K_{z\delta }+\frac{\lambda_{35}K_{my\delta }}{K_{J}}\right)\cdot \delta_{r} \end{array}$$(7)$$\begin{array}{cc} \dot{\beta }= & \frac{1}{A_{x}v_{A}}\left(K_{z\beta }v_{A}^{2}-\frac{\lambda_{35}K_{my\beta }v_{A}^{2}}{K_{J}}-T\right)\cdot \beta \\ \; & \;+\frac{1}{A_{x}}\left(A_{x}+K_{z\omega }-\lambda_{33}-\frac{\lambda_{35}K_{my\omega }-\lambda_{35}^{2}}{K_{J}}\right)\cdot \omega_{y}\\ \; & \;+\frac{v_{A}}{A_{x}}\left(K_{z\delta }-\frac{\lambda_{35}K_{my\delta }}{K_{J}}\right)\cdot \delta_{r} \end{array}$$(8)$$\begin{array}{cc} {\dot{\omega }}_{y}= & \frac{v_{A}^{2}}{K_{J}}\left[K_{my\delta }+\frac{\lambda_{35}}{A_{x}}\left(-K_{z\delta }+\frac{\lambda_{35}K_{my\delta }}{K_{J}}\right)\right]\cdot \delta_{r}\\ \; & +\frac{1}{K_{J}}\left[K_{my\beta }v_{A}^{2}-\frac{\lambda_{35}}{A_{x}}\left(K_{z\beta }v_{A}^{2}-\frac{\lambda_{35}K_{my\beta }v_{A}^{2}}{K_{J}}-T\right)\right]\cdot \beta \\ \; & +\frac{v_{A}}{K_{J}}\left[K_{my\omega }-\lambda_{35}\right]\cdot \omega_{y}+\frac{v_{A}}{K_{J}}\left[\frac{\lambda_{35}}{A_{x}}\left(-K_{z\omega }+\lambda_{33}+\frac{\lambda_{35}K_{my\omega }-\lambda_{35}^{2}}{K_{J}}\right)\right]\cdot \omega_{y} \end{array}$$

In the above, $\sigma_{A}=\psi_{A}-\beta$, where $\psi_{A}$ is the yaw and β is the sideslip angle; $\omega_{y}$ is the yaw velocity and $\delta_{r}$ is the rudder control input to be designed. Furthermore, the constant *m* is the mass of the AUV; the constant $K_{J}$ is the moment of inertia, the constants $\lambda_{33}\;\mathrm{and}\;\lambda_{35}$ are the hydrodynamic coefficients, the constant $A_{x}≜\frac{\left(m+\lambda_{33}\right)K_{J}-\lambda_{35}^{2}}{K_{J}}$, the constants $K_{z\beta },K_{z\delta },K_{z\omega },K_{my\delta },K_{my\beta },K_{my\omega }$ are the correlation coefficients, and the constant *T* is the end trajectory thrust in the longitudinal direction of the AUV.

Substituting (Equation 6) into (Equation 5) yields(9)$$\begin{array}{cc} r\ddot{q}= & -2\dot{r}\frac{x_{1}}{r}-\frac{1}{A_{x}}\left(-K_{z\beta }v_{A}^{2}+\frac{\lambda_{35}K_{my\beta }v_{A}^{2}}{K_{J}}+T\right)\cdot \beta \\ \; & \;-\frac{v_{A}}{A_{x}}\left(-K_{z\omega }+\lambda_{33}+\frac{\lambda_{35}K_{my\omega }-\lambda_{35}^{2}}{K_{J}}\right)\cdot \omega_{y}\;-\frac{v_{A}^{2}}{A_{x}}\left(-K_{z\delta }+\frac{\lambda_{35}K_{my\delta }}{K_{J}}\right)\cdot \delta_{r}+\Delta_{q} \end{array}$$
where $x_{1}≜r\dot{q}$, and the derivative of $x_{1}$ along (Equation 9) gives(10)$$\begin{array}{cc} {\dot{x}}_{1}= & \dot{r}\dot{q}+r\ddot{q}=\frac{\dot{r}}{r}x_{1}+r\ddot{q}\\ = & -\frac{\dot{r}}{r}x_{1}+\frac{1}{A_{x}}\left(K_{z\beta }v_{A}^{2}-\frac{\lambda_{35}K_{my\beta }v_{A}^{2}}{K_{J}}-T\right)\cdot \beta \\ \; & \;+\frac{v_{A}}{A_{x}}\left(K_{z\omega }-\lambda_{33}-\frac{\lambda_{35}K_{my\omega }-\lambda_{35}^{2}}{K_{J}}\right)\cdot \omega_{y}\;+\frac{v_{A}^{2}}{A_{x}}\left(K_{z\delta }-\frac{\lambda_{35}K_{my\delta }}{K_{J}}\right)\cdot \delta_{r}+\Delta_{q} \end{array}$$

Define the state variables $x_{2}≜\beta$ and $x_{3}≜\omega_{y}$, and combining Equations (Equation 7), (Equation 8) and (Equation 10), the IGC model of the AUV is given by(11)$$\dot{x}=Ax+B\delta_{r}+\Delta$$
where $x={\left[x_{1},x_{2},x_{3}\right]}^{⊤}$, $A=\left(\begin{array}{ccc} a_{11} & a_{12} & a_{13}\\ 0 & a_{22} & a_{23}\\ 0 & a_{32} & a_{33} \end{array}\right)$, $B={\left[b_{1},b_{2},b_{3}\right]}^{⊤}$, $\Delta ={\left[\Delta_{q},0,0\right]}^{⊤}$, $a_{11}=-\frac{\dot{r}}{r}$, $a_{12}=\frac{1}{A_{x}}\left(K_{z\beta }v_{A}^{2}-\frac{\lambda_{35}K_{my\beta }v_{A}^{2}}{K_{J}}-T\right)$, $a_{13}=\frac{v_{A}}{A_{x}}\left(K_{z\omega }-\lambda_{33}-\frac{\lambda_{35}K_{my\omega }-\lambda_{35}^{2}}{K_{J}}\right)$, $b_{1}=\frac{v_{A}^{2}}{A_{x}}\left(K_{z\delta }-\frac{\lambda_{35}K_{my\delta }}{K_{J}}\right)$; $a_{22}=\frac{a_{12}}{v_{A}}$, $a_{23}=1+\frac{a_{13}}{v_{A}},b_{2}=\frac{b_{1}}{v_{A}}$; $a_{32}=\frac{K_{my\beta }v_{A}^{2}}{K_{J}}-\frac{\lambda_{35}}{K_{J}}a_{12}$, $a_{33}=\frac{v_{A}}{K_{J}}\left(K_{my\omega }-\lambda_{35}-\frac{\lambda_{35}}{v_{A}}a_{13}\right)$, $b_{3}=\frac{v_{A}^{2}}{K_{J}}\left(K_{my\delta }-\frac{\lambda_{35}}{v_{A}}b_{2}\right)$.

It is then obtained from (Equation 2) that(12)$${\dot{a}}_{12}={\dot{a}}_{13}={\dot{a}}_{22}={\dot{a}}_{23}={\dot{a}}_{32}={\dot{a}}_{33}=0$$

Define $\bar{x}={\left[x_{0},\;x^{⊤}\right]}^{⊤}$ and $x_{0}≜rq$; using (Equation 11), we obtain(13)$$\left\{\begin{array}{c} \dot{\bar{x}}=\bar{A}\bar{x}+B\delta_{r}+\Delta \\ y={\left[x_{0},\;x_{3}\right]}^{⊤}≜{\left[rq,\;\omega_{y}\right]}^{⊤} \end{array}\right.$$
where$$\bar{A}=\left[\begin{array}{cccc} 0 & 1 & 0 & 0\\ 0 & a_{11} & a_{12} & a_{13}\\ 0 & 0 & a_{22} & a_{23}\\ 0 & 0 & a_{32} & a_{33} \end{array}\right],\;\bar{B}=\left[\begin{array}{c} 0\\ b_{1}\\ b_{2}\\ b_{3} \end{array}\right],\;\bar{\Delta }=\left[\begin{array}{c} \dot{r}q\\ \Delta_{q}\\ 0\\ 0 \end{array}\right],$$
and y is the system output vector measured by the equipped acoustic transducer array and inertial navigation system.

After the above derivation, the control objective reduces to enable the design of the rudder control input $\delta_{r}$ with the available system outputs y, i.e., $\dot{q}$ and β are unmeasurable, such that the state vector $\bar{x}$ is stabilized.

## 3. Main Results

Considering the modeling error and external disturbances in the system, (Equation 13) is rewritten as(14)$$\begin{array}{cc} \; & {\dot{x}}_{0}=x_{1}+\dot{r}q\\ \; & {\dot{x}}_{1}=a_{11}x_{1}+a_{12}x_{2}+a_{13}x_{3}+b_{1}\delta_{r}+d_{1}\\ \; & {\dot{x}}_{2}=a_{22}x_{2}+a_{23}x_{3}+b_{2}\delta_{r}+d_{2}\\ \; & {\dot{x}}_{3}=a_{32}x_{2}+a_{33}x_{3}+b_{3}\delta_{r}+d_{3} \end{array}$$
where $d_{1}≜\Delta_{q}+\Delta_{1}$, $\Delta_{1},d_{2},d_{3}$ are unknown nonlinearities, including modeling error and the external disturbances of each system state.

### 3.1. Model Transformation

It is noted from (Equation 14) that the LOS velocity $\dot{q}$ and the sideslip angle β are not available, leading to unknown $x_{1}$ and $x_{2}$. Therefore, an observer needs to be designed to achieve the online estimation of $x_{1}$ and $x_{2}$. Since $d_{1}$, $d_{2}$, and $d_{3}$ are unmatched disturbances in the system, to facilitate the design of the observer and controller, a novel state transformation method is proposed.

The primary objective of this transformation is to convert the original system affected by unmatched disturbances into an equivalent system with matched disturbances. Specifically, the proposed transformation enables the development of an observer capable of accurately estimating the unknown states, $x_{1}$ and $x_{2}$, and the resultant external disturbances, thereby providing the necessary information for the IGC design. The subsequent sections of this paper will detail the formulation of this transformation and its integration into the overall IGC framework.

Using the measurable yaw velocity $x_{3}$, the input transformation is given by(15)$$\delta_{r}=\frac{u-a_{13}x_{3}}{b_{1}}$$

Substituting (Equation 15) into (Equation 14) yields(16)$$\begin{array}{cc} \; & {\dot{x}}_{0}=x_{1}+\dot{r}q\\ \; & {\dot{x}}_{1}=a_{11}x_{1}+a_{12}x_{2}+u+d_{1}\\ \; & {\dot{x}}_{2}=a_{22}x_{2}+{\bar{a}}_{23}x_{3}+\frac{b_{2}}{b_{1}}u+d_{2}\\ \; & {\dot{x}}_{3}=a_{32}x_{2}+{\bar{a}}_{33}x_{3}+\frac{b_{3}}{b_{1}}u+d_{3} \end{array}$$
where ${\bar{a}}_{23}≜a_{23}-\frac{b_{2}}{b_{1}}a_{13}=1$, ${\bar{a}}_{33}≜a_{33}-\frac{b_{3}}{b_{1}}a_{13}$.

It is observed from (Equation 16) that the unknown disturbances exist in the last three equations. By lumping the disturbances with system states, the state transformation is given by(17)$$\begin{array}{cc} z_{0}= & x_{0}\\ z_{1}= & x_{1}\\ z_{2}= & a_{12}x_{2}+d_{1}\\ z_{3}= & a_{12}{\bar{a}}_{23}x_{3}+a_{12}d_{2}+{\dot{d}}_{1}-a_{22}d_{1}\\ z_{4}= & a_{12}{\bar{a}}_{23}d_{3}+a_{12}{\dot{d}}_{2}+{\ddot{d}}_{1}-\left(a_{22}+{\bar{a}}_{33}\right){\dot{d}}_{1}\\ \; & +\left(a_{22}{\bar{a}}_{33}-{\bar{a}}_{23}a_{32}\right)d_{1}-a_{12}{\bar{a}}_{33}d_{2} \end{array}$$

Applying (Equation 17), the first two equations of (Equation 16) become(18)$$\begin{array}{cc} \; & {\dot{z}}_{0}=z_{1}+\dot{r}q\\ \; & {\dot{z}}_{1}=a_{11}z_{1}+u+z_{2} \end{array}$$

Substituting (Equation 16) and $a_{12}x_{2}=z_{2}-d_{1}$ of (Equation 17) into ${\dot{z}}_{2}$ with (Equation 12), we have(19)$$\begin{array}{cc} {\dot{z}}_{2}= & a_{12}{\dot{x}}_{2}+{\dot{d}}_{1}=a_{12}\left(a_{22}x_{2}+{\bar{a}}_{23}x_{3}+\frac{b_{2}}{b_{1}}u+d_{2}\right)+{\dot{d}}_{1}\\ \; & =a_{22}\left(z_{2}-d_{1}\right)+a_{12}{\bar{a}}_{23}x_{3}+\frac{a_{12}b_{2}}{b_{1}}u+a_{12}d_{2}+{\dot{d}}_{1}\\ \; & =a_{22}z_{2}+\frac{a_{12}b_{2}}{b_{1}}u+a_{12}{\bar{a}}_{23}x_{3}+a_{12}d_{2}+{\dot{d}}_{1}-a_{22}d_{1}\\ \; & =a_{22}z_{2}+\frac{a_{12}b_{2}}{b_{1}}u+z_{3} \end{array}$$

Substituting the last equation of (Equation 16) into $\dot{z_{3}}$ yields(20)$$\begin{array}{cc} {\dot{z}}_{3}= & a_{12}{\bar{a}}_{23}\left(a_{32}x_{2}+{\bar{a}}_{33}x_{3}+\frac{b_{3}}{b_{1}}u+d_{3}\right)+a_{12}{\dot{d}}_{2}+{\ddot{d}}_{1}-a_{22}{\dot{d}}_{1}\\ = & a_{12}{\bar{a}}_{23}a_{32}x_{2}+a_{12}{\bar{a}}_{23}{\bar{a}}_{33}x_{3}+\frac{a_{12}{\bar{a}}_{23}b_{3}}{b_{1}}u+a_{12}{\bar{a}}_{23}d_{3}+a_{12}{\dot{d}}_{2}+{\ddot{d}}_{1}-a_{22}{\dot{d}}_{1} \end{array}$$

According to (Equation 17), we obtain $a_{12}x_{2}=z_{2}-d_{1}$ and $a_{12}{\bar{a}}_{23}x_{3}=z_{3}-a_{12}d_{2}-{\dot{d}}_{1}+a_{22}d_{1}$, and then (Equation 20) becomes(21)$$\begin{array}{cc} {\dot{z}}_{3}= & {\bar{a}}_{23}a_{32}\left(z_{2}-d_{1}\right)+{\bar{a}}_{33}\left(z_{3}-a_{12}d_{2}-{\dot{d}}_{1}+a_{22}d_{1}\right)\\ \; & +\frac{a_{12}{\bar{a}}_{23}b_{3}}{b_{1}}u+a_{12}{\bar{a}}_{23}d_{3}+a_{12}{\dot{d}}_{2}+{\ddot{d}}_{1}-a_{22}{\dot{d}}_{1}\\ = & {\bar{a}}_{23}a_{32}z_{2}+{\bar{a}}_{33}z_{3}+\frac{a_{12}{\bar{a}}_{23}b_{3}}{b_{1}}u+d_{L} \end{array}$$
where the lumped disturbance of the transformed system is given by$$\begin{array}{cc} d_{L}= & a_{12}{\bar{a}}_{23}d_{3}+a_{12}{\dot{d}}_{2}+{\ddot{d}}_{1}-a_{22}{\dot{d}}_{1}-{\bar{a}}_{23}a_{32}d_{1}-a_{12}{\bar{a}}_{33}d_{2}-{\bar{a}}_{33}{\dot{d}}_{1}+a_{22}{\bar{a}}_{33}d_{1} \end{array}$$

Using the input transformation (Equation 15) and the state transformation (Equation 17), the system (Equation 14) with unmatched disturbances can be transformed as(22)$$\begin{array}{cc} \; & {\dot{z}}_{0}=z_{1}+\dot{r}q\\ \; & {\dot{z}}_{1}=z_{2}+a_{11}z_{1}+u\\ \; & {\dot{z}}_{2}=z_{3}+a_{22}z_{2}+{\bar{b}}_{2}u\\ \; & {\dot{z}}_{3}=z_{4}+{\bar{a}}_{32}z_{2}+{\bar{a}}_{33}z_{3}+{\bar{b}}_{3}u \end{array}$$
where $z_{4}≜d_{L}$, ${\bar{a}}_{32}={\bar{a}}_{23}a_{32}$, ${\bar{b}}_{2}=\frac{b_{2}}{b_{1}}a_{12}$, ${\bar{b}}_{3}=\frac{b_{3}}{b_{1}}a_{12}{\bar{a}}_{23}$.

Accordingly, the control objective reduces to enable the designing of *u* to stabilize the states of (Equation 22) with unknown $z_{1}$, $z_{2}$, $z_{3}$, and $z_{4}$. The actual rudder control input $\delta_{r}$ can be obtained by using (Equation 15) and (Equation 17). The IGC design diagram is shown in Figure 2.

In order to facilitate the derivation and analysis of the ESO, the following assumptions, definitions, and lemmas are given.

**Assumption** **1.**
*The lumped disturbance $d_{L}$ and its derivative ${\dot{d}}_{L}$ are bounded, such that there exists a known constant $\bar{\xi }>0$ satisfying $\left|{\dot{d}}_{L}\right|\leq \bar{\xi }$.*


**Remark** **1.**
*Since the system modeling omitted the higher-order small quantities, and the target has limited maneuvering ability, it is determined that $\cos(q-\sigma_{A})\approx 1$; therefore, Assumption 1 is reasonable.*


**Assumption** **2.**
*The value of the sideslip angle β is very small during the interception trajectory. It is assumed that sin(β)≈β, cos(β)≈1.*


**Assumption** **3.**
*Using Assumptions 1 and 2, it is determined that the disturbed sideslip angle signal $z_{2}$ is bounded, and its estimation $v_{2}$ can be bounded a priori by the ESO design. It is consequently assumed that there exists a known constant w>0 satisfying ${\bar{a}}_{32}\left|e_{2}\right|\leq w$ with the estimated error $e_{2}=z_{2}-v_{2}$.*


**Definition** **1**([41,42])**.** *Consider the following system:*(23)$$\dot{x}=f\left(x\right),\;\;x\in U\subseteq R^{n},\;\;f\left(0\right)=0$$
*where $f:U\to R^{n}$ is a continuous function of x on an open neighborhood $U\subseteq R^{n}$ of the origin. $\forall x_{0}\in U_{0}\subset R^{n}$, there exists a continuous function $T\left(x\right)$$\;:\;U_{0}\setminus \left\{0\right\}\to \left(0,\;+\infty \right)$, such that the solution $x\left(t,\;x_{0}\right)$ of (Equation 23) satisfies $x\left(t,\;x_{0}\right)\in U_{0}\setminus \left\{0\right\}$ and $\lim_{t\to T\left(x_{0}\right)}\;x\left(t,\;x_{0}\right)=0$, $\forall t\in \left[0,\;T\left(x_{0}\right)\right)$; and $x\left(t,\;x_{0}\right)=0$, $\forall t>T\left(x_{0}\right)$; then, x will converge from $x_{0}$ to 0 in finite time $T\left(x_{0}\right)$.*

**Lemma** **1**([41])**.** *Consider the system described by (Equation 23); if there exist continuously differentiable positive definite functions $V:U\to R^{n}$, positive real numbers c>0 and $\alpha \in \left(0,\;1\right)$, and the open neighborhood $U_{0}\subset U$ containing the origin, such that*$$\dot{V}\left(x\right)+c{\left(V\left(x\right)\right)}^{\alpha }\leq 0,\;\;x\in U_{0}\setminus \left\{0\right\}$$
*then it is determined that the system (Equation 23) is finite-time stable. In addition, the system (Equation 23) is globally finite-time stable with the convergence time, satisfying $t\leq t_{0}+\frac{1}{c\left(1-\alpha \right)}{\left(V{\left(x\right)}_{t=t_{0}}\right)}^{\left(1-\alpha \right)}$ if $U=U_{0}\subseteq R^{n}$ and $V\left(x\right)$ is radially unbounded.*

**Lemma** **2**([43,44])**.** *Consider the system described by (Equation 23); if there exist positive real numbers $c_{1},\;c_{2}>0$, $\alpha \in \left(0,\;1\right)$, the continuously differentiable positive definite function $V:U\to R^{n}$, and the open neighborhood $U_{0}\subset U$ that contains the origin, such that*$$\dot{V}\left(x\right)+c_{1}V\left(x\right)+c_{2}{\left(V\left(x\right)\right)}^{\alpha }\leq 0,\;\;x\in U_{0}\setminus \left\{0\right\}$$
*then the system (Equation 23) is finite-time stable.*
*If $U=U_{0}\subseteq R^{n}$ and $V\left(x\right)$ is radially unbounded, then the system (Equation 23) is globally finite-time stable with the convergence time, satisfying*

$$t\leq t_{0}+\frac{1}{c_{1}\left(1-\alpha \right)}\cdot ln\frac{c_{1}\cdot {\left(V{\left(x\right)}_{t=t_{0}}\right)}^{\left(1-\alpha \right)}+c_{2}}{c_{2}}$$

*In addition, if there exists ϵ∈R, 0<ϵ<∞ such that*

$$\dot{V}\left(x\right)+c_{1}V\left(x\right)+c_{2}{\left(V\left(x\right)\right)}^{\alpha }\leq \epsilon$$

*then (Equation 23) is finite-time stable and the states converge to*

$$U=\left\{\left.x\;\right|V\left(x\right)\leq min\left\{\frac{\epsilon }{\left(1-c_{0}\right)c_{1}},\;{\left(\frac{\epsilon }{\left(1-c_{0}\right)c_{2}}\right)}^{\alpha }\right\}\right\}$$

*where $0<c_{0}<1$, and the convergence time satisfies*

$$\begin{array}{cc} t\leq & max\left\{t_{0}+\frac{1}{c_{0}c_{1}\left(1-\alpha \right)}\cdot ln\frac{c_{0}c_{1}\cdot {\left(V{\left(x\right)}_{t=t_{0}}\right)}^{\left(1-\alpha \right)}+c_{2}}{c_{2}},\right.\\ \; & \;t_{0}+\left.\frac{1}{c_{1}\left(1-\alpha \right)}\cdot ln\frac{c_{1}\cdot {\left(V{\left(x\right)}_{t=t_{0}}\right)}^{\left(1-\alpha \right)}+c_{0}c_{2}}{c_{0}c_{2}}\right\} \end{array}$$



**Definition** **2**([45,46])**.** *Define the function vector $f\left(x\right)={\left[f_{1}\left(x\right),\;f_{2}\left(x\right),\;\cdots ,\;f_{n}\left(x\right)\right]}^{⊤}:R^{n}\to R^{n}$. For any ε>0, there exist a constant vector $\left[r_{1},\;r_{2},\;\cdots ,\;r_{n}\right]\in R^{n}$, $r_{i}>0$, i=1,2,⋯,n, and a constant $k>-min\left\{r_{i},\;i=1,\;2,\;\cdots ,\;n\right\}$, such that*$$f_{i}\left(\varepsilon^{r_{1}}x_{1},\;\varepsilon^{r_{2}}x_{2},\;\cdots ,\;\varepsilon^{r_{n}}x_{n}\right)=\varepsilon^{\left(k+r_{i}\right)}\cdot f_{i}\left(x\right),\;$$
*then f(x) is said to have a homogeneous degree k with respect to the dilation $\left[r_{1},\;r_{2},\;\cdots ,\;r_{n}\right]$.*

**Definition** **3**([45,46])**.** *Define a continuous scalar function $V\left(x\right):R^{n}\to R$, for any ε>0, as follows: if there exist k>0 and the dilation $\left[r_{1},\;r_{2},\;\cdots ,\;r_{n}\right]\in R^{n}$, $r_{i}>0$, i=1,2,⋯,n, such that*$$V\left(\varepsilon^{r_{1}}x_{1},\;\varepsilon^{r_{2}}x_{2},\;\cdots ,\;\varepsilon^{r_{n}}x_{n}\right)=\varepsilon^{k}\cdot V\left(x\right),\;\forall x\in R^{n},$$
*then $V\left(x\right)$ is said to have a homogeneous degree k with respect to the dilation $\left[r_{1},\;r_{2},\;\cdots ,\;r_{n}\right]$.*

**Lemma** **3**([47,48])**.** *The system (Equation 23) is globally finite-time stable, if (Equation 23) is globally asymptotically stable and has a negative homogeneous degree k<0.*

**Lemma** **4**([49])**.** *For any c>0, a>1, b>1, x,y∈R, if $\left(a-1\right)\left(b-1\right)=1,\;$ the following relationship holds:*$$xy\leq \frac{c^{a}}{a}{\left|x\right|}^{a}+\frac{1}{bc^{b}}{\left|y\right|}^{b}$$

**Lemma** **5**([50])**.** *For any x,y∈R, ε>0, 0<α<1, the following relationship holds:*$$-x{\left|x+y\right|}^{\alpha }sign\left(x+y\right)\leq -{\left|x\right|}^{\left(\alpha +1\right)}+\frac{\varepsilon }{2}x^{2}+\frac{2}{\varepsilon }y^{2\alpha }$$

### 3.2. Finite-Time ESO Design

In this section, a finite-time ESO is designed to quickly estimate the unknown states of the system (Equation 22) with matched disturbances, by which the rudder control input $\delta_{r}$ is designed. It is noted from (Equation 22) that the system states $z_{i},\;i=1,\;2,\;3,\;4$ are unknown due to the disturbances ($d_{i}$) and unmeasurable variables ($\dot{q}$, β), and only $z_{0}$ is a known signal among the state variables. Since $z_{4}≜d_{L}$ needs to be extended as a new state for the ESO design, it follows from (Equation 21) that the extended system state satisfies ${\dot{z}}_{4}={\dot{d}}_{L}$. Thus, (Equation 22) is rewritten as(24)$$\begin{array}{cc} \; & {\dot{z}}_{0}=z_{1}+\dot{r}q\\ \; & {\dot{z}}_{1}=z_{2}+a_{11}z_{1}+u\\ \; & {\dot{z}}_{2}=z_{3}+a_{22}z_{2}+{\bar{b}}_{2}u\\ \; & {\dot{z}}_{3}=z_{4}+{\bar{a}}_{32}z_{2}+{\bar{a}}_{33}z_{3}+{\bar{b}}_{3}u\\ \; & {\dot{z}}_{4}={\dot{d}}_{L}≜\xi \left(t\right) \end{array}$$

Let $v_{i},\;i=0,\;1,\;2,\;3,\;4$, denote the estimations of $z_{i}$ in (Equation 24); then, the finite-time ESO is designed as(25)$$\begin{array}{cc} {\dot{v}}_{0}= & v_{1}+\dot{r}q+\rho_{0}{\left|e_{0}\right|}^{\alpha_{0}}\mathrm{sign}\left(e_{0}\right)+\rho_{0}{\left|e_{0}\right|}^{\beta_{0}}\mathrm{sign}\left(e_{0}\right)+k_{0}\mathrm{sign}\left(e_{0}\right)\\ {\dot{v}}_{1}= & v_{2}+a_{11}v_{1}+u+\rho_{1}{\left|e_{0}\right|}^{\alpha_{1}}\mathrm{sign}\left(e_{0}\right)+\rho_{1}{\left|e_{0}\right|}^{\beta_{1}}\mathrm{sign}\left(e_{0}\right)+k_{1}\mathrm{sign}\left(e_{0}\right)\\ {\dot{v}}_{2}= & v_{3}+a_{22}v_{2}+{\bar{b}}_{2}u+\rho_{2}{\left|e_{0}\right|}^{\alpha_{2}}\mathrm{sign}\left(e_{0}\right)+\rho_{2}{\left|e_{0}\right|}^{\beta_{2}}\mathrm{sign}\left(e_{0}\right)+k_{2}\mathrm{sign}\left(e_{0}\right)\\ {\dot{v}}_{3}= & v_{4}+{\bar{a}}_{32}v_{2}+{\bar{a}}_{33}v_{3}+{\bar{b}}_{3}u+\rho_{3}{\left|e_{0}\right|}^{\alpha_{3}}\mathrm{sign}\left(e_{0}\right)+\rho_{3}{\left|e_{0}\right|}^{\beta_{3}}\mathrm{sign}\left(e_{0}\right)+k_{3}\mathrm{sign}\left(e_{0}\right)\\ {\dot{v}}_{4}= & \rho_{4}{\left|e_{0}\right|}^{\alpha_{4}}\mathrm{sign}\left(e_{0}\right)+\rho_{4}{\left|e_{0}\right|}^{\beta_{4}}\mathrm{sign}\left(e_{0}\right)+k_{4}\mathrm{sign}\left(e_{0}\right) \end{array}$$
where $k_{i}>0$, $\rho_{i}>1$, $\alpha_{i}=\left(i+1\right)\alpha_{0}-i$, $\beta_{i}=\beta_{0}+i\left(\alpha_{0}-1\right)$, i=0,1,2,3,4, and $\alpha_{0}$ and $\beta_{0}$ are selected such that $0.8<\alpha_{0}<1$, $\beta_{0}=1/\alpha_{0}$.

Let $e_{i}=z_{i}-v_{i}$ denote the estimation error; then, applying (Equation 24) and (Equation 25) yields(26)$$\begin{array}{cc} {\dot{e}}_{0}= & e_{1}-\rho_{0}{\left|e_{0}\right|}^{\alpha_{0}}\mathrm{sign}\left(e_{0}\right)-\rho_{0}{\left|e_{0}\right|}^{\beta_{0}}\mathrm{sign}\left(e_{0}\right)-k_{0}\mathrm{sign}\left(e_{0}\right)\\ {\dot{e}}_{1}= & e_{2}+a_{11}e_{1}-\rho_{1}{\left|e_{0}\right|}^{\alpha_{1}}\mathrm{sign}\left(e_{0}\right)-\rho_{1}{\left|e_{0}\right|}^{\beta_{1}}\mathrm{sign}\left(e_{0}\right)-k_{1}\mathrm{sign}\left(e_{0}\right)\\ {\dot{e}}_{2}= & e_{3}+a_{22}e_{2}-\rho_{2}{\left|e_{0}\right|}^{\alpha_{2}}\mathrm{sign}\left(e_{0}\right)-\rho_{2}{\left|e_{0}\right|}^{\beta_{2}}\mathrm{sign}\left(e_{0}\right)-k_{2}\mathrm{sign}\left(e_{0}\right)\\ {\dot{e}}_{3}= & e_{4}+{\bar{a}}_{32}e_{2}+{\bar{a}}_{33}e_{3}-\rho_{3}{\left|e_{0}\right|}^{\alpha_{3}}\mathrm{sign}\left(e_{0}\right)-\rho_{3}{\left|e_{0}\right|}^{\beta_{3}}\mathrm{sign}\left(e_{0}\right)-k_{3}\mathrm{sign}\left(e_{0}\right)\\ {\dot{e}}_{4}= & -\rho_{4}{\left|e_{0}\right|}^{\alpha_{4}}\mathrm{sign}\left(e_{0}\right)-\rho_{4}{\left|e_{0}\right|}^{\beta_{4}}\mathrm{sign}\left(e_{0}\right)-k_{4}\mathrm{sign}\left(e_{0}\right)+\xi \left(t\right) \end{array}$$

Let $e={\left[e_{0},\;e_{1},\;e_{2},\;e_{3},\;e_{4}\right]}^{⊤}$ denote the estimation error vector of the ESO, by which (Equation 26) equals to(27)$$\dot{e}=f_{\alpha }+f_{\beta }+f_{\sigma }$$
where the vector fields are given by(28)$$f_{\alpha }=\left[\begin{array}{c} e_{1}-\rho_{0}{\left|e_{0}\right|}^{\alpha_{0}}\mathrm{sign}\left(e_{0}\right)\\ e_{2}-\rho_{1}{\left|e_{0}\right|}^{\alpha_{1}}\mathrm{sign}\left(e_{0}\right)\\ e_{3}-\rho_{2}{\left|e_{0}\right|}^{\alpha_{2}}\mathrm{sign}\left(e_{0}\right)\\ e_{4}-\rho_{3}{\left|e_{0}\right|}^{\alpha_{3}}\mathrm{sign}\left(e_{0}\right)\\ -\rho_{4}{\left|e_{0}\right|}^{\alpha_{4}}\mathrm{sign}\left(e_{0}\right) \end{array}\right],\;f_{\beta }=\left[\begin{array}{c} -\rho_{0}{\left|e_{0}\right|}^{\beta_{0}}\mathrm{sign}\left(e_{0}\right)\\ -\rho_{1}{\left|e_{0}\right|}^{\beta_{1}}\mathrm{sign}\left(e_{0}\right)\\ -\rho_{2}{\left|e_{0}\right|}^{\beta_{2}}\mathrm{sign}\left(e_{0}\right)\\ -\rho_{3}{\left|e_{0}\right|}^{\beta_{3}}\mathrm{sign}\left(e_{0}\right)\\ -\rho_{4}{\left|e_{0}\right|}^{\beta_{4}}\mathrm{sign}\left(e_{0}\right) \end{array}\right]$$(29)$$f_{\sigma }=\left[\begin{array}{c} -k_{0}\mathrm{sign}\left(e_{0}\right)\\ a_{11}e_{1}-k_{1}\mathrm{sign}\left(e_{0}\right)\\ a_{22}e_{2}-k_{2}\mathrm{sign}\left(e_{0}\right)\\ {\bar{a}}_{32}e_{2}+{\bar{a}}_{33}e_{3}-k_{3}\mathrm{sign}\left(e_{0}\right)\\ -k_{4}\mathrm{sign}\left(e_{0}\right)+\xi \left(t\right) \end{array}\right]$$

Define the constant $\vartheta ≜\alpha_{0}\cdot \alpha_{1}\cdot \alpha_{2}\cdot \alpha_{3}>0$ and the error vector(30)$$\bar{e}=\left[\begin{array}{c} {\left|e_{0}\right|}^{\frac{1}{\vartheta }}\cdot \mathrm{sign}\left(e_{0}\right)\\ {\left|e_{1}\right|}^{\frac{1}{\alpha_{0}\vartheta }}\cdot \mathrm{sign}\left(e_{1}\right)\\ {\left|e_{2}\right|}^{\frac{1}{\alpha_{1}\vartheta }}\cdot \mathrm{sign}\left(e_{2}\right)\\ {\left|e_{3}\right|}^{\frac{1}{\alpha_{2}\vartheta }}\cdot \mathrm{sign}\left(e_{3}\right)\\ {\left|e_{4}\right|}^{\frac{1}{\alpha_{3}\vartheta }}\cdot \mathrm{sign}\left(e_{4}\right)\; \end{array}\right]$$

Construct the Lyapunov function(31)$$V={\bar{e}}^{⊤}P\bar{e}$$
where the symmetric positive-definite matrix P can be obtained by solving the following equation:(32)$$A_{p}^{⊤}P+PA_{p}=-I_{5\times 5}$$
where I is the unit diagonal matrix; the design parameters $k_{i}>0$, i=0,1,2,3,4 of (Equation 26) are selected such that (Equation 33) is Hurwitz.(33)$$A_{p}=\left[\begin{array}{ccccc} -k_{0} & 1 & 0 & 0 & 0\\ -k_{1} & 0 & 1 & 0 & 0\\ -k_{2} & 0 & 0 & 1 & 0\\ -k_{3} & 0 & 0 & 0 & 1\\ -k_{4} & 0 & 0 & 0 & 0 \end{array}\right]$$

The convergence analysis of the ESO given by (Equation 25) is summarized in *Theorem 1*.

**Theorem** **1.**
*Consider the transformed AUV IGC model (Equation 14) satisfying Assumptions 1–3; the design parameters are selected as $k_{i}>0$, $\rho_{i}>1$, $\alpha_{i}=\left(i+1\right)\alpha_{0}-i$, $\beta_{i}=\beta_{0}+i\left(\alpha_{0}-1\right)$, i=0,1,2,3,4; $0.8<\alpha_{0}<1$, $\beta_{0}=1/\alpha_{0}$. Then, the estimation error $e_{i}=z_{i}-v_{i}$ of the finite-time ESO (Equation 25) will converge to the set U defined by (Equation 34) in finite time $T_{0}≜t_{1}+t_{2}<\infty$.*

(34)
$$U=\left\{\left.e\;\right|∥e∥\leq \sum_{k=0}^{4}\Omega_{k}\right\}$$

*where ∀j=1,2,3,4,*

$$\begin{array}{c} \Omega_{0}=\frac{1}{{\left(\sqrt{\lambda_{min}\left(P\right)}\right)}^{\vartheta }}{\left(\frac{\sum_{k=4}^{8}\mu_{k}}{\mu_{1}\left(1-\mu_{0}\right)-\mu_{3}}\right)}^{\frac{1}{\alpha_{0}}},\;\Omega_{j}=\frac{1}{{\left(\sqrt{\lambda_{min}\left(P\right)}\right)}^{\alpha_{j-1}\cdot \vartheta }}{\left(\frac{\sum_{k=4}^{8}\mu_{k}}{\mu_{1}\left(1-\mu_{0}\right)-\mu_{3}}\right)}^{\frac{\alpha_{j-1}\cdot \vartheta }{\alpha_{0}\vartheta }},\;\\ 0<\mu_{0}<1-\sum_{k=3}^{8}\frac{\mu_{k}}{\mu_{1}},\;\;\mu_{1}=-\underset{\left\{y:\;V_{\alpha }\left(y\right)=1\right\}}{max}\;L_{f_{\alpha }}V_{\alpha }\left(y\right),\;\mu_{2}=-\underset{\left\{y:\;V_{\beta }\left(y\right)=1\right\}}{max}\;L_{f_{\beta }}V_{\beta }\left(y\right),\;\\ \mu_{3}=\frac{2\lambda_{max}\left(P\right)}{\lambda_{min}\left(P\right)}\left(\frac{a_{11}}{\alpha_{0}\vartheta }+\frac{a_{22}}{\alpha_{1}\vartheta }+\frac{{\bar{a}}_{33}}{\alpha_{2}\vartheta }\right),\;\mu_{4}=\frac{2k_{0}\lambda_{max}\left(P\right)}{\vartheta \lambda_{min}\left(P\right)},\;\mu_{5}=\frac{2k_{1}\lambda_{max}\left(P\right)}{\alpha_{0}\vartheta \lambda_{min}\left(P\right)},\;\\ \mu_{6}=\frac{2k_{2}\lambda_{max}\left(P\right)}{\alpha_{1}\vartheta \lambda_{min}\left(P\right)},\;\mu_{7}=\frac{2\left(w+k_{3}\right)\lambda_{max}\left(P\right)}{\alpha_{2}\vartheta \lambda_{min}\left(P\right)},\;\mu_{8}=\frac{2\left(\bar{\xi }+k_{4}\right)\lambda_{max}\left(P\right)}{\alpha_{3}\vartheta \lambda_{min}\left(P\right)},\; \end{array}$$

*where $L_{f_{\alpha }}V$ is the Lie derivative of V with respect to the vector field $f_{\alpha }$, i.e., the time derivative of V along the system:*

(35)
$$\dot{e}=f_{\alpha }$$

*where $L_{f_{\beta }}V$ is the Lie derivative of V with respect to vector field $f_{\beta }$, i.e., the time derivative of V along the system:*

(36)
$$\dot{e}=f_{\beta }$$


*In addition, the convergence time $t_{1},\;t_{2}$ satisfies*

(37)
$$\begin{array}{cc} t_{1}< & \frac{{\left(V{\left(\bar{e}\right)}_{t=0}\right)}^{1-\gamma_{1}}}{\mu_{1}\left(1-\gamma_{1}\right)}\cdot G\left(1,\;\frac{1-\gamma_{1}}{\gamma_{2}-\gamma_{1}},\right.\;\left.1+\frac{1-\gamma_{1}}{\gamma_{2}-\gamma_{1}},\;-\frac{{\bar{\mu }}_{2}}{\mu_{1}}{\left(V{\left(\bar{e}\right)}_{t=0}\right)}^{\gamma_{2}-\gamma_{1}}\right) \end{array}$$


(38)
$$\begin{array}{cc} t_{2}< & \frac{{\left(V{\left(\bar{e}\right)}_{t=t_{1}}\right)}^{1-\gamma_{1}}}{\mu_{1}\mu_{0}\left(1-\gamma_{1}\right)}\cdot G\left(1,\;\frac{1-\gamma_{1}}{\gamma_{2}-\gamma_{1}},\;\right.\left.1+\frac{1-\gamma_{1}}{\gamma_{2}-\gamma_{1}},\;-\frac{\mu_{2}}{\mu_{1}\mu_{0}}\right) \end{array}$$

*where ${\bar{\mu }}_{2}≜\mu_{2}-\sum_{k=3}^{8}\mu_{k}$, $\gamma_{1}=1+\vartheta \alpha_{0}/2-\vartheta /2$, $\gamma_{2}=1+\vartheta /\left(2\alpha_{0}\right)-\vartheta /2$, and $G\left(\cdot \right)$ is the Gaussian hypergeometric function [51,52].*


**Proof.** The derivative of (Equation 31) along (Equation 27) is obtained as(39)$$\begin{array}{c} \dot{V}={\dot{\bar{e}}}^{⊤}P\bar{e}+{\bar{e}}^{⊤}P\dot{\bar{e}}=L_{f_{\alpha }}V+L_{f_{\beta }}V+\Theta \end{array}$$
where $\Theta ≜L_{f_{\sigma }}V$, satisfying(40)$$\Theta =2{\bar{e}}^{⊤}P\left[\begin{array}{c} -{\left|e_{0}\right|}^{\left(\frac{1}{\vartheta }-1\right)}\cdot \frac{k_{0}\mathrm{sign}\left(e_{0}\right)}{\vartheta }\\ {\left|e_{1}\right|}^{\left(\frac{1}{\alpha_{0}\vartheta }-1\right)}\cdot \frac{a_{11}e_{1}-k_{1}\mathrm{sign}\left(e_{0}\right)}{\alpha_{0}\vartheta }\\ {\left|e_{2}\right|}^{\left(\frac{1}{\alpha_{1}\vartheta }-1\right)}\cdot \frac{a_{22}e_{2}-k_{2}\mathrm{sign}\left(e_{0}\right)}{\alpha_{1}\vartheta }\\ {\left|e_{3}\right|}^{\left(\frac{1}{\alpha_{2}\vartheta }-1\right)}\cdot \frac{{\bar{a}}_{32}e_{2}+{\bar{a}}_{33}e_{3}-k_{3}\mathrm{sign}\left(e_{0}\right)}{\alpha_{2}\vartheta }\\ {\left|e_{4}\right|}^{\left(\frac{1}{\alpha_{3}\vartheta }-1\right)}\cdot \frac{\xi \left(t\right)-k_{4}\mathrm{sign}\left(e_{0}\right)}{\alpha_{3}\vartheta } \end{array}\right]$$
in view of (Equation 29) and (Equation 30).The result of (Equation 39) shows that $\dot{V}$ along (Equation 27) can be decomposed into the sum of the Lie derivatives of *V* with respect to the vector fields $f_{\alpha }$, $f_{\beta }$, and $f_{\sigma }$, such that the three Lie derivatives on the right side of (Equation 39) can be studied separately.For $L_{f_{\alpha }}V$, combining (Equation 28) and expanding (Equation 35) yield(41)$$\begin{array}{cc} \; & {\dot{e}}_{0}=e_{1}-\rho_{0}{\left|e_{0}\right|}^{\alpha_{0}}\mathrm{sign}\left(e_{0}\right)\\ \; & {\dot{e}}_{1}=e_{2}-\rho_{1}{\left|e_{0}\right|}^{\alpha_{1}}\mathrm{sign}\left(e_{0}\right)\\ \; & {\dot{e}}_{2}=e_{3}-\rho_{2}{\left|e_{0}\right|}^{\alpha_{2}}\mathrm{sign}\left(e_{0}\right)\\ \; & {\dot{e}}_{3}=e_{4}-\rho_{3}{\left|e_{0}\right|}^{\alpha_{3}}\mathrm{sign}\left(e_{0}\right)\\ \; & {\dot{e}}_{4}=-\rho_{4}{\left|e_{0}\right|}^{\alpha_{4}}\mathrm{sign}\left(e_{0}\right) \end{array}$$
Construct the Lyapunov function as follows:(42)$$V_{\alpha }={\bar{e}}^{⊤}P\bar{e}$$
Notice that $V_{\alpha }$ has the same expression as *V* in (Equation 31), while the Lie derivatives are along different systems. The subscript α emphasizes that $V_{\alpha }$ is a Lyapunov function of the system $\dot{e}=f_{\alpha }$ in (Equation 35) and (Equation 41).Taking the time-derivative of (Equation 42) along (Equation 35) yields(43)$${\dot{V}}_{\alpha }=L_{f_{\alpha }}V_{\alpha }=L_{f_{\alpha }}V$$According to [46], the system defined by (Equation 35) and (Equation 41) has a negative degree of homogeneity $\alpha_{0}-1<0$ with respect to the dilation $\left[1,\;\alpha_{0},\;\alpha_{1},\;\alpha_{2},\;\alpha_{3}\right]$. It is then verified from *Definition 3* that the scalar functions $V_{\alpha }$ and $L_{f_{\alpha }}V_{\alpha }$ have degrees of homogeneity 2/ϑ and $2/\vartheta +\alpha_{0}-1$ with respect to the dilation $\left[1,\;\alpha_{0},\;\alpha_{1},\;\alpha_{2},\;\alpha_{3}\right]$; using Lemma 4.2 of [48] yields(44)$$L_{f_{\alpha }}V=L_{f_{\alpha }}V_{\alpha }\leq -\mu_{1}V_{\alpha }^{\gamma_{1}}=-\mu_{1}V^{\gamma_{1}}$$
where $\mu_{1}=-\max_{\left\{y:\;V_{\alpha }\left(y\right)=1\right\}}\;L_{f_{\alpha }}V_{\alpha }\left(y\right)$, $\gamma_{1}=\frac{2/\vartheta +\alpha_{0}-1}{2/\vartheta }=1+\vartheta \alpha_{0}/2-\vartheta /2<1$.Furthermore, according to Theorem 1 of [53], we have(45)$$\max_{\alpha_{0}\to 1}\;\mu_{1}\geq \frac{\rho }{\lambda_{\max}\left(P\right)}$$
where $\rho =\min\left\{\rho_{i},\;i=0,\;1,\;2,\;3,\;4\right\}>1$ is the minimum value of the ESO design parameter $\rho_{i}$ given by (Equation 25), and $\lambda_{\max}\left(P\right)>0$ is the largest eigenvalue of the positive-definite matrix P.Similarly to the analytical steps of (Equation 41) to (Equation 45), for $L_{f_{\beta }}V$, combining (Equation 28) and expanding (Equation 36) yield(46)$$\begin{array}{cc} \; & {\dot{e}}_{0}=-\rho_{0}{\left|e_{0}\right|}^{\beta_{0}}\mathrm{sign}\left(e_{0}\right)\\ \; & {\dot{e}}_{1}=-\rho_{1}{\left|e_{0}\right|}^{\beta_{1}}\mathrm{sign}\left(e_{0}\right)\\ \; & {\dot{e}}_{2}=-\rho_{2}{\left|e_{0}\right|}^{\beta_{2}}\mathrm{sign}\left(e_{0}\right)\\ \; & {\dot{e}}_{3}=-\rho_{3}{\left|e_{0}\right|}^{\beta_{3}}\mathrm{sign}\left(e_{0}\right)\\ \; & {\dot{e}}_{4}=-\rho_{4}{\left|e_{0}\right|}^{\beta_{4}}\mathrm{sign}\left(e_{0}\right) \end{array}$$Define the Lyapunov function(47)$$V_{\beta }={\bar{e}}^{⊤}P\bar{e}$$
Notice that $V_{\beta }$ has the same expression as *V* in (Equation 31); the subscript β emphasizes that $V_{\beta }$ is a Lyapunov function of $\dot{e}=f_{\beta }$ in (Equation 36) and (Equation 46).Taking the time-derivative of (Equation 47) along (Equation 36) yields(48)$${\dot{V}}_{\beta }=L_{f_{\beta }}V_{\beta }=L_{f_{\beta }}V$$According to [46], the system defined by (Equation 36) and (Equation 46) has a negative degree of homogeneity $\beta_{0}-1$ with respect to the dilation $\left[1,\;\alpha_{0},\;\alpha_{1},\;\alpha_{2},\;\alpha_{3}\right]$. It is then verified from *Definition 3* that the scalar functions $V_{\beta }$ and $L_{f_{\beta }}V_{\beta }$ have degrees of homogeneity 2/ϑ and $2/\vartheta +\beta_{0}-1$ with respect to the dilation $\left[1,\;\alpha_{0},\;\alpha_{1},\;\alpha_{2},\;\alpha_{3}\right]$; using Lemma 4.2 of [48] yields(49)$$L_{f_{\beta }}V=L_{f_{\beta }}V_{\beta }\leq -\mu_{2}V_{\beta }^{\gamma_{2}}=-\mu_{2}V^{\gamma_{2}}$$
where $\mu_{2}=-\max_{\left\{y:\;V_{\beta }\left(y\right)=1\right\}}\;L_{f_{\beta }}V_{\beta }\left(y\right)$, $\gamma_{2}=\frac{2/\vartheta +\beta_{0}-1}{2/\vartheta }=1+\vartheta /\left(2\alpha_{0}\right)-\vartheta /2>1$.Furthermore, according to Theorem 1 of [53], we have(50)$$\max_{\alpha_{0}\to 1}\;\mu_{2}\geq \frac{\rho }{\lambda_{\max}\left(P\right)}$$
where ρ and $\lambda_{\max}\left(P\right)$ are defined in (Equation 45).For $\Theta ≜L_{f_{\sigma }}V$, substituting (Equation 44) and (Equation 49) into (Equation 39) yields(51)$$\dot{V}\leq -\mu_{1}V^{\gamma_{1}}-\mu_{2}V^{\gamma_{2}}+\Theta$$Combining (Equation 40) with *Assumptions 1, 2, and 3*, we obtain(52)$$\begin{array}{cc} \Theta \leq & ∥\bar{e}∥\cdot \left({\left|e_{0}\right|}^{\left(\frac{1}{\vartheta }-1\right)}\cdot \frac{k_{0}}{\vartheta }\right.+{\left|e_{1}\right|}^{\left(\frac{1}{\alpha_{0}\vartheta }-1\right)}\cdot \frac{a_{11}\left|e_{1}\right|+k_{1}}{\alpha_{0}\vartheta }+{\left|e_{2}\right|}^{\left(\frac{1}{\alpha_{1}\vartheta }-1\right)}\cdot \frac{a_{22}\left|e_{2}\right|+k_{2}}{\alpha_{1}\vartheta }\\ \; & +{\left|e_{3}\right|}^{\left(\frac{1}{\alpha_{2}\vartheta }-1\right)}\cdot \frac{w+{\bar{a}}_{33}\left|e_{3}\right|+k_{3}}{\alpha_{2}\vartheta }\left.+{\left|e_{4}\right|}^{\left(\frac{1}{\alpha_{3}\vartheta }-1\right)}\cdot \frac{\bar{\xi }+k_{4}}{\alpha_{3}\vartheta }\right)\cdot 2\lambda_{\max}\left(P\right) \end{array}$$It can be seen from (Equation 30) that(53)$$\begin{array}{c} {\left|e_{0}\right|}^{\frac{1}{\vartheta }}\leq ∥\bar{e}∥\\ {\left|e_{j+1}\right|}^{\frac{1}{\alpha_{j}\vartheta }}\leq ∥\bar{e}∥,\;\forall j=0,\;1,\;2,\;3 \end{array}$$Thus, (Equation 53) is equivalent to(54)$$\begin{array}{c} \begin{array}{cc} \left|e_{0}\right| & \leq {∥\bar{e}∥}^{\vartheta }\\ \left|e_{k}\right| & \leq {∥\bar{e}∥}^{\alpha_{k-1}\cdot \vartheta },\;\forall k=1,\;2,\;3,\;4\\ {\left|e_{0}\right|}^{\left(\frac{1}{\vartheta }-1\right)} & ={\left|e_{0}\right|}^{\frac{1}{\vartheta }\cdot \left(1-\vartheta \right)}\leq {∥\bar{e}∥}^{\left(1-\vartheta \right)}\\ {\left|e_{j+1}\right|}^{\left(\frac{1}{\alpha_{j}\vartheta }-1\right)} & ={\left|e_{j+1}\right|}^{\frac{1}{\alpha_{j}\vartheta }\cdot \left(1-\alpha_{j}\vartheta \right)}\leq {∥\bar{e}∥}^{\left(1-\alpha_{j}\vartheta \right)},\;\forall j=0,\;1,\;2,\;3 \end{array} \end{array}$$Substituting (Equation 53) and (Equation 54) into (Equation 52) yields(55)$$\begin{array}{cc} \Theta \leq & 2\lambda_{\max}\left(P\right)\cdot \left(\frac{k_{0}}{\vartheta }{∥\bar{e}∥}^{\left(2-\vartheta \right)}+\frac{a_{11}}{\alpha_{0}\vartheta }{∥\bar{e}∥}^{2}\right.+\frac{{\bar{a}}_{33}}{\alpha_{2}\vartheta }{∥\bar{e}∥}^{2}+\frac{a_{22}}{\alpha_{1}\vartheta }{∥\bar{e}∥}^{2}+\frac{k_{2}}{\alpha_{1}\vartheta }{∥\bar{e}∥}^{\left(2-\alpha_{1}\vartheta \right)}\\ \; & +\frac{k_{1}}{\alpha_{0}\vartheta }{∥\bar{e}∥}^{\left(2-\alpha_{0}\vartheta \right)}+\frac{w+k_{3}}{\alpha_{2}\vartheta }{∥\bar{e}∥}^{\left(2-\alpha_{2}\vartheta \right)}+\left.\frac{\bar{\xi }+k_{4}}{\alpha_{3}\vartheta }{∥\bar{e}∥}^{\left(2-\alpha_{3}\vartheta \right)}\right) \end{array}$$Let $\lambda_{\min}\left(P\right)>0$ denote the minimum eigenvalue of the positive-definite matrix P; we have(56)$$\lambda_{\min}\left(P\right){∥\bar{e}∥}^{2}\leq V,\;\;{∥\bar{e}∥}^{2}\leq \frac{V}{\lambda_{\min}\left(P\right)}$$Substituting (Equation 56) and (Equation 55) into (Equation 51) yields(57)$$\begin{array}{cc} \dot{V}\leq & -\mu_{1}V^{\gamma_{1}}-\mu_{2}V^{\gamma_{2}}+\mu_{3}V+\mu_{4}V^{\left(1-\vartheta /2\right)}+\mu_{5}V^{\left(1-\alpha_{0}\vartheta /2\right)}\\ \; & +\mu_{6}V^{\left(1-\alpha_{1}\vartheta /2\right)}+\mu_{7}V^{\left(1-\alpha_{2}\vartheta /2\right)}+\mu_{8}V^{\left(1-\alpha_{3}\vartheta /2\right)} \end{array}$$
where $\mu_{3}=\frac{2\lambda_{\max}\left(P\right)}{\lambda_{\min}\left(P\right)}\left(\frac{a_{11}}{\alpha_{0}\vartheta }+\frac{a_{22}}{\alpha_{1}\vartheta }+\frac{{\bar{a}}_{33}}{\alpha_{2}\vartheta }\right)$, $\mu_{4}=\frac{2k_{0}\lambda_{\max}\left(P\right)}{\vartheta \lambda_{\min}\left(P\right)}$, $\mu_{5}=\frac{2k_{1}\lambda_{\max}\left(P\right)}{\alpha_{0}\vartheta \lambda_{\min}\left(P\right)}$, $\mu_{6}=\frac{2k_{2}\lambda_{\max}\left(P\right)}{\alpha_{1}\vartheta \lambda_{\min}\left(P\right)}$, $\mu_{7}=\frac{2\left(w+k_{3}\right)\lambda_{\max}\left(P\right)}{\alpha_{2}\vartheta \lambda_{\min}\left(P\right)}$, $\mu_{8}=\frac{2\left(\bar{\xi }+k_{4}\right)\lambda_{\max}\left(P\right)}{\alpha_{3}\vartheta \lambda_{\min}\left(P\right)}$.**Remark** **2.**
*It should be noted that $\dot{r}$ is bounded because the AUV and the target have limited velocities. Additionally, the interception concludes as $\left|r\right|<r_{min}$, which means $\left|r\right|>r_{min}$ during the interception; therefore, $a_{11}=-\frac{\dot{r}}{r}$ is bounded [54].*
Since $0.8<\alpha_{0}<1$, using (Equation 44) and (Equation 49) yields(58)$$\begin{array}{c} \gamma_{1}<1,\\ 1-\frac{\vartheta }{2}<1-\frac{\alpha_{j}\vartheta }{2}<1<\gamma_{2},\;\forall j=0,\;1,\;2,\;3 \end{array}$$When V≥1, using (Equation 57) and (Equation 58), we have ∀V≥1,(59)$$\;\dot{V}\leq -\mu_{1}V^{\gamma_{1}}-\left(\mu_{2}-\sum_{k=3}^{8}\mu_{k}\right)V^{\gamma_{2}}$$Since $\lim_{\alpha_{0}\to 1}\;\mu_{2}\geq \frac{\rho }{\lambda_{\max}\left(P\right)}$, select the design parameters $\rho_{i},\;i=0,\;1,\;2,\;3,\;4$, to satisfy(60)$${\bar{\mu }}_{2}≜\mu_{2}-\sum_{k=3}^{8}\mu_{k}>0$$
such that(61)$$\dot{V}\leq -\mu_{1}V^{\gamma_{1}}-{\bar{\mu }}_{2}V^{\gamma_{2}}\leq 0,\;\;\forall V\geq 1$$Therefore, the Lyapunov function *V* can converge from any initial value $V{\left(\bar{e}\right)}_{t=0}$ to V≡1 in finite-time $t\in \left[0,\;t_{1}\right)$, satisfying(62)$$\begin{array}{cc} t_{1}< & \frac{{\left(V{\left(\bar{e}\right)}_{t=0}\right)}^{1-\gamma_{1}}}{\mu_{1}\left(1-\gamma_{1}\right)}\cdot G\left(1,\;\frac{1-\gamma_{1}}{\gamma_{2}-\gamma_{1}},\;\right.\left.1+\frac{1-\gamma_{1}}{\gamma_{2}-\gamma_{1}},\;-\frac{{\bar{\mu }}_{2}}{\mu_{1}}{\left(V{\left(\bar{e}\right)}_{t=0}\right)}^{\gamma_{2}-\gamma_{1}}\right) \end{array}$$When V<1, using (Equation 57) and (Equation 58), we have ∀V<1,(63)$$\dot{V}\leq -\mu_{1}V^{\gamma_{1}}-\mu_{2}V^{\gamma_{2}}+\mu_{3}V^{\gamma_{1}}+\sum_{k=4}^{8}\mu_{k}V^{\left(1-\vartheta /2\right)}$$Since $\lim_{\alpha_{0}\to 1}\;\mu_{1}\geq \frac{\rho }{\lambda_{\max}\left(P\right)}$, select the design parameters $\rho_{i},\;i=0,\;1,\;2,\;3,\;4$, to satisfy $\mu_{1}-\sum_{k=3}^{8}\mu_{k}>0$, and define the constant(64)$$0<\mu_{0}<1-\sum_{k=3}^{8}\frac{\mu_{k}}{\mu_{1}}$$Applying (Equation 64) into (Equation 63) yields(65)$$\begin{array}{cc} \dot{V}\leq & -\mu_{1}\left(1-\mu_{0}\right)V^{\gamma_{1}}-\mu_{0}V^{\gamma_{1}}-\mu_{2}V^{\gamma_{2}}+\mu_{3}V^{\gamma_{1}}+\sum_{k=4}^{8}\mu_{k}V^{\left(1-\vartheta /2\right)}\\ = & -\left(\mu_{1}\left(1-\mu_{0}\right)-\mu_{3}\right)V^{\gamma_{1}}+\sum_{k=4}^{8}\mu_{k}V^{\left(1-\vartheta /2\right)}-\mu_{1}\mu_{0}V^{\gamma_{1}}-\mu_{2}V^{\gamma_{2}}\\ \leq & -\left[\left(\mu_{1}\left(1-\mu_{0}\right)-\mu_{3}\right)V^{\left(\gamma_{1}-1+\vartheta /2\right)}\right.\;\;\;\;\;\left.-\sum_{k=4}^{8}\mu_{k}\right]V^{\left(1-\vartheta /2\right)}-\mu_{1}\mu_{0}V^{\gamma_{1}}-\mu_{2}V^{\gamma_{2}} \end{array}$$According to (Equation 65), it is easy to verify that $\dot{V}<0$, ∀V<1 if (Equation 66) holds.(66)$$\left(\mu_{1}\left(1-\mu_{0}\right)-\mu_{3}\right)V^{\left(\gamma_{1}-1+\vartheta /2\right)}-\sum_{k=4}^{8}\mu_{k}>0$$Therefore, the Lyapunov function *V* can converge from the initial value $V{\left(\bar{e}\right)}_{t=t_{1}}=1$ to the set(67)$$V<{\left(\frac{\sum_{k=4}^{8}\mu_{k}}{\mu_{1}\left(1-\mu_{0}\right)-\mu_{3}}\right)}^{\frac{2}{2\gamma_{1}-2+\vartheta }}$$
in finite time $t\in \left[t_{1},\;t_{1}+t_{2}\right)$, satisfying(68)$$\begin{array}{cc} t_{2}< & \frac{{\left(V{\left(\bar{e}\right)}_{t=t_{1}}\right)}^{1-\gamma_{1}}}{\mu_{1}\mu_{0}\left(1-\gamma_{1}\right)}\cdot G\left(1,\;\frac{1-\gamma_{1}}{\gamma_{2}-\gamma_{1}},\;\right.\left.1+\frac{1-\gamma_{1}}{\gamma_{2}-\gamma_{1}},\;-\frac{\mu_{2}}{\mu_{1}\mu_{0}}\right) \end{array}$$The above analysis shows that *V* converges from the set defined by (Equation 67) in finite time $t\leq T_{0}≜t_{1}+t_{2}<\infty$. By applying (Equation 56) and (Equation 67), we obtain that the observer error vector $\bar{e}$ satisfies(69)$$∥\bar{e}∥\leq \frac{1}{\sqrt{\lambda_{\min}\left(P\right)}}{\left(\frac{\sum_{k=4}^{8}\mu_{k}}{\mu_{1}\left(1-\mu_{0}\right)-\mu_{3}}\right)}^{\frac{1}{2\gamma_{1}-2+\vartheta }}$$Note that $2\gamma_{1}-2+\vartheta =\alpha_{0}\vartheta$. Using (Equation 54), we can determine that the observation error vector $e={\left[e_{0},\;e_{1},\;e_{2},\;e_{3},\;e_{4}\right]}^{⊤}$ can converge to the set *U* defined by (Equation 34) in finite time $T_{0}$. Once again, using (Equation 45) and (Equation 50), it is seen that $\mu_{1}$ and $\mu_{2}$ can be increased by adjusting $\rho_{i}$, such that *U* is made arbitrarily small. □

*Theorem 1* shows that by adjusting the design parameters $\rho_{i},\;i=0,\;1,\;2,\;3,\;4$, the estimation error vector can converge in finite time within the residual set *U* defined by (Equation 34). It is further shown that when the design parameters $k_{i}$ are selected to be large enough, the observation error vector can converge to 0 in finite time, which is summarized in *Theorem 2*.

**Theorem** **2.**
*Consider the AUV IGC model (Equation 14) satisfying Assumptions 1–3, and the design parameters are selected according to Theorem 1; if the design parameters $k_{i},\;i=0,\;1,\;2,\;3,\;4$ further satisfy $k_{0}>\Omega_{1}$, $k_{1}>\left|a_{11}\right|\Omega_{1}+\Omega_{2}$, $k_{2}>\left|a_{22}\right|\Omega_{2}+\Omega_{3}$, $k_{3}>\left|{\bar{a}}_{33}\right|\Omega_{3}+\Omega_{4}$, $k_{4}>\bar{\xi }$, where the specific definitions of $\Omega_{i}$ are shown in (Equation 34), then the observation error vector e of the ESO (Equation 25) will converge to **0** in finite time $t\leq \sum_{k=1}^{7}t_{k}$, and the convergence time $t_{j},\;j=3,\;4,\;5,\;6,\;7$ satisfy*

(70)
$$\begin{array}{cc} t_{3} & <\frac{{\left(V_{0}{\left(e_{0}\right)}_{t=T_{0}}\right)}^{\frac{1-\alpha_{0}}{2}}}{2^{\frac{\alpha_{0}-1}{2}}\rho_{0}\left(1-\alpha_{0}\right)}\cdot G\left(1,\;\frac{1-\alpha_{0}}{\beta_{0}-\alpha_{0}},\;\right.\left.1+\frac{1-\alpha_{0}}{\beta_{0}-\alpha_{0}},\;-2^{\frac{\beta_{0}-\alpha_{0}}{2}}{\left(V_{0}{\left(e_{0}\right)}_{t=T_{0}}\right)}^{\frac{\beta_{0}-\alpha_{0}}{2}}\right) \end{array}$$


(71)
$$t_{4}\leq \frac{\sqrt{2}}{{\bar{k}}_{1}}{\left(V_{1}{\left(e_{1}\right)}_{t=T_{1}}\right)}^{\frac{1}{2}}$$


(72)
$$t_{5}\leq \frac{\sqrt{2}}{{\bar{k}}_{2}}{\left(V_{2}{\left(e_{2}\right)}_{t=T_{2}}\right)}^{\frac{1}{2}}$$


(73)
$$t_{6}\leq \frac{\sqrt{2}}{{\bar{k}}_{3}}{\left(V_{3}{\left(e_{3}\right)}_{t=T_{3}}\right)}^{\frac{1}{2}}$$


(74)
$$t_{7}\leq \frac{\sqrt{2}}{{\bar{k}}_{4}}{\left(V_{4}{\left(e_{4}\right)}_{t=T_{4}}\right)}^{\frac{1}{2}}$$



**Proof.** Define the Lyapunov function $V_{0}=\frac{1}{2}e_{0}^{2}$, and taking the time-derivative of $V_{0}$ along (Equation 26) yields(75)$$\begin{array}{cc} {\dot{V}}_{0} & =e_{0}{\dot{e}}_{0}=e_{0}e_{1}-\rho_{0}{\left|e_{0}\right|}^{\left(\alpha_{0}+1\right)}-\rho_{0}{\left|e_{0}\right|}^{\left(\beta_{0}+1\right)}-k_{0}\left|e_{0}\right|\\ \; & \leq -\rho_{0}{\left|e_{0}\right|}^{\left(\alpha_{0}+1\right)}-\rho_{0}{\left|e_{0}\right|}^{\left(\beta_{0}+1\right)}-\left(k_{0}-\left|e_{1}\right|\right)\left|e_{0}\right| \end{array}$$According to (Equation 54), (Equation 69), and (Equation 34), we have $\left|e_{1}\right|\leq {∥\bar{e}∥}^{\alpha_{0}\vartheta }\leq \Omega_{1}$. It is obtained that ${\dot{V}}_{0}\leq 0$ in (Equation 75) by choosing the design parameter(76)$$k_{0}>\Omega_{1},$$
by which (Equation 75) is rewritten as(77)$${\dot{V}}_{0}\leq -2^{\frac{\alpha_{0}+1}{2}}\rho_{0}\cdot {V_{0}}^{\frac{\alpha_{0}+1}{2}}-2^{\frac{\beta_{0}+1}{2}}\rho_{0}\cdot {V_{0}}^{\frac{\beta_{0}+1}{2}}$$
with $\left|e_{0}\right|={\left(2V_{0}\right)}^{\frac{1}{2}}$.Therefore, $V_{0}$ can converge from the initial value $V_{0}{\left(e_{0}\right)}_{t=T_{0}}$ to zero in finite-time $t\in \left[T_{0},\;T_{0}+t_{3}\right)$ satisfying(78)$$\begin{array}{cc} t_{3}< & \frac{{\left(V_{0}{\left(e_{0}\right)}_{t=T_{0}}\right)}^{\frac{1-\alpha_{0}}{2}}}{2^{\frac{\alpha_{0}-1}{2}}\rho_{0}\left(1-\alpha_{0}\right)}\cdot G\left(1,\;\frac{1-\alpha_{0}}{\beta_{0}-\alpha_{0}},\right.\;\;\;\;\left.1+\frac{1-\alpha_{0}}{\beta_{0}-\alpha_{0}},\;-2^{\frac{\beta_{0}-\alpha_{0}}{2}}{\left(V_{0}{\left(e_{0}\right)}_{t=T_{0}}\right)}^{\frac{\beta_{0}-\alpha_{0}}{2}}\right) \end{array}$$The analysis of (Equation 77) and (Equation 78) implies that ${\dot{e}}_{0}=e_{0}\equiv 0$ in finite-time $t\leq T_{1}≜T_{0}+t_{3}<\infty$, such that the first equation of (Equation 26) is rewritten as(79)$$e_{1}={\left(\rho_{0}{\left|e_{0}\right|}^{\alpha_{0}}\mathrm{sign}\left(e_{0}\right)+\rho_{0}{\left|e_{0}\right|}^{\beta_{0}}\mathrm{sign}\left(e_{0}\right)+k_{0}\mathrm{sign}\left(e_{0}\right)\right)}_{\mathrm{eq}}$$
which is obtained by passing the signal$$\rho_{0}{\left|e_{0}\right|}^{\alpha_{0}}\mathrm{sign}\left(e_{0}\right)+\rho_{0}{\left|e_{0}\right|}^{\beta_{0}}\mathrm{sign}\left(e_{0}\right)+k_{0}\mathrm{sign}\left(e_{0}\right)$$
through a low-pass filter based on the equivalent control theory [55], i.e., the low-frequency part of the sign function is preserved.From (Equation 79) we obtain(80)$$\begin{array}{cc} \mathrm{sign}\left(e_{1}\right)=\mathrm{sign} & \left(\rho_{0}{\left|e_{0}\right|}^{\alpha_{0}}\mathrm{sign}\left(e_{0}\right)\right.\;\;+\rho_{0}{\left|e_{0}\right|}^{\beta_{0}}\mathrm{sign}\left(e_{0}\right)+k_{0}\mathrm{sign}\left(e_{0}\right)) \end{array}$$
therefore we have $\mathrm{sign}\left(e_{1}\right)=\mathrm{sign}\left(e_{0}\right)$, and applying ${\dot{e}}_{0}=e_{0}\equiv 0$ into the second equation of (Equation 26) yields(81)$${\dot{e}}_{1}=e_{2}+a_{11}e_{1}-k_{1}\mathrm{sign}\left(e_{1}\right)$$Define the Lyapunov function $V_{1}=\frac{1}{2}e_{1}^{2}$, and taking the time-derivative of $V_{1}$ along (Equation 81) yields(82)$$\begin{array}{cc} {\dot{V}}_{1}=e_{1}{\dot{e}}_{1}= & e_{1}e_{2}+a_{11}e_{1}^{2}-k_{1}\left|e_{1}\right|\leq -\left(k_{1}-\left|a_{11}e_{1}\right|-\left|e_{2}\right|\right)\left|e_{1}\right| \end{array}$$According to (Equation 54), (Equation 69) and (Equation 34), we have $\left|e_{1}\right|\leq {∥\bar{e}∥}^{\alpha_{0}\vartheta }\leq \Omega_{1}$, $\left|e_{2}\right|\leq {∥\bar{e}∥}^{\alpha_{1}\vartheta }\leq \Omega_{2}$. It is observed that ${\dot{V}}_{1}\leq 0$ in (Equation 82) by choosing the design parameter(83)$$k_{1}>\left|a_{11}\right|\Omega_{1}+\Omega_{2},$$
by which (Equation 82) is rewritten as(84)$${\dot{V}}_{1}\leq -\sqrt{2}{\bar{k}}_{1}{V_{1}}^{\frac{1}{2}}$$
where ${\bar{k}}_{1}≜k_{1}-\left|a_{11}e_{1}\right|-\left|e_{2}\right|$ and $\left|e_{1}\right|={\left(2V_{1}\right)}^{\frac{1}{2}}$.Therefore, by combining (Equation 84) with *Lemma 1*, we obtain that the Lyapunov function $V_{1}$ can converge from the initial value $V_{1}{\left(e_{1}\right)}_{t=T_{1}}$ to zero in finite time $t\in \left[T_{1},\;T_{1}+t_{4}\right)$, satisfying(85)$$t_{4}\leq \frac{\sqrt{2}}{{\bar{k}}_{1}}{\left(V_{1}{\left(e_{1}\right)}_{t=T_{1}}\right)}^{\frac{1}{2}}$$The analysis of (Equation 84) and (Equation 85) implies that ${\dot{e}}_{1}=e_{1}\equiv 0$ in finite time $t\leq T_{2}≜T_{1}+t_{4}<\infty$, such that (Equation 81) is rewritten as(86)$$e_{2}={\left(k_{1}\mathrm{sign}\left(e_{1}\right)\right)}_{\mathrm{eq}}$$Similar to the analytical steps of (Equation 79) to (Equation 81), we obtain $\mathrm{sign}\left(e_{2}\right)=\mathrm{sign}\left(e_{0}\right)$, such that the third equation of (Equation 26) equals to(87)$${\dot{e}}_{2}=e_{3}+a_{22}e_{2}-k_{2}\mathrm{sign}\left(e_{2}\right)$$Define the Lyapunov function $V_{2}=\frac{1}{2}e_{2}^{2}$, and taking the time-derivative of $V_{2}$ along (Equation 87) yields(88)$$\begin{array}{c} {\dot{V}}_{2}=e_{2}{\dot{e}}_{2}=e_{2}e_{3}+a_{22}e_{2}^{2}-k_{2}\left|e_{2}\right|\leq -\left(k_{2}-\left|a_{22}e_{2}\right|-\left|e_{3}\right|\right)\left|e_{2}\right| \end{array}$$According to (Equation 54), (Equation 69), and (Equation 34), we have $\left|e_{3}\right|\leq {∥\bar{e}∥}^{\alpha_{2}\vartheta }\leq \Omega_{3}$. It is observed that ${\dot{V}}_{2}\leq 0$ in (Equation 88) by choosing the design parameter(89)$$k_{2}>\left|a_{22}\right|\Omega_{2}+\Omega_{3}$$
by which (Equation 88) is rewritten as(90)$${\dot{V}}_{2}\leq -\sqrt{2}{\bar{k}}_{2}{V_{2}}^{\frac{1}{2}}$$
where ${\bar{k}}_{2}≜k_{2}-\left|a_{22}e_{2}\right|-\left|e_{3}\right|$ and $\left|e_{2}\right|={\left(2V_{2}\right)}^{\frac{1}{2}}$.Therefore, by combining (Equation 90) with *Lemma 1*, we obtain that the Lyapunov function $V_{2}$ can converge from the initial value $V_{2}{\left(e_{2}\right)}_{t=T_{2}}$ to zero in finite time $t\in \left[T_{2},\;T_{2}+t_{5}\right)$, satisfying(91)$$t_{5}\leq \frac{\sqrt{2}}{{\bar{k}}_{2}}{\left(V_{2}{\left(e_{2}\right)}_{t=T_{2}}\right)}^{\frac{1}{2}}$$The analysis of (Equation 90) and (Equation 91) implies that ${\dot{e}}_{2}=e_{2}\equiv 0$ in finite time $t\leq T_{3}≜T_{2}+t_{5}<\infty$, such that (Equation 87) is rewritten as(92)$$e_{3}={\left(k_{2}\mathrm{sign}\left(e_{2}\right)\right)}_{\mathrm{eq}}$$Similarly to the analytical steps of (Equation 79) to (Equation 81), we obtain $\mathrm{sign}\left(e_{3}\right)=\mathrm{sign}\left(e_{0}\right)$, such that the fourth equation of (Equation 26) equals to(93)$${\dot{e}}_{3}=e_{4}+{\bar{a}}_{33}e_{3}-k_{3}\mathrm{sign}\left(e_{3}\right)$$Define the Lyapunov function $V_{3}=\frac{1}{2}e_{3}^{2}$, and taking the time-derivative of $V_{3}$ along (Equation 93) yields(94)$$\begin{array}{c} {\dot{V}}_{3}=e_{3}{\dot{e}}_{3}=e_{3}e_{4}+{\bar{a}}_{33}e_{3}^{2}-k_{3}\left|e_{3}\right|\leq -\left(k_{3}-\left|{\bar{a}}_{33}e_{3}\right|-\left|e_{4}\right|\right)\left|e_{3}\right| \end{array}$$According to (Equation 54), (Equation 69) and (Equation 34), we have $\left|e_{4}\right|\leq {∥\bar{e}∥}^{\alpha_{3}\vartheta }\leq \Omega_{4}$. It is observed that ${\dot{V}}_{3}\leq 0$ in (Equation 94) by choosing the design parameter(95)$$k_{3}>\left|{\bar{a}}_{33}\right|\Omega_{3}+\Omega_{4}$$
by which (Equation 94) is rewritten as(96)$${\dot{V}}_{3}\leq -\sqrt{2}{\bar{k}}_{3}{V_{3}}^{\frac{1}{2}}$$
where ${\bar{k}}_{3}≜k_{3}-\left|{\bar{a}}_{33}e_{3}\right|-\left|e_{4}\right|$ and $\left|e_{3}\right|={\left(2V_{3}\right)}^{\frac{1}{2}}$.Therefore, by combining (Equation 96) with *Lemma 1*, we obtain that the Lyapunov function $V_{3}$ can converge from the initial value $V_{3}{\left(e_{3}\right)}_{t=T_{3}}$ to zero in finite time $t\in \left[T_{3},\;T_{3}+t_{6}\right)$, satisfying(97)$$t_{6}\leq \frac{\sqrt{2}}{{\bar{k}}_{3}}{\left(V_{3}{\left(e_{3}\right)}_{t=T_{3}}\right)}^{\frac{1}{2}}$$The analysis of (Equation 96) and (Equation 97) implies that ${\dot{e}}_{3}=e_{3}\equiv 0$ in finite time $t\leq T_{4}≜T_{3}+t_{6}<\infty$, such that (Equation 93) is rewritten as(98)$$e_{4}={\left(k_{3}\mathrm{sign}\left(e_{3}\right)\right)}_{\mathrm{eq}}$$Similarly to the analytical steps of (Equation 79) to (Equation 81), we obtain $\mathrm{sign}\left(e_{4}\right)=\mathrm{sign}\left(e_{0}\right)$, such that the last equation of (Equation 26) equals to(99)$${\dot{e}}_{4}=-k_{4}\mathrm{sign}\left(e_{4}\right)+\xi \left(t\right)$$Define the Lyapunov function $V_{4}=\frac{1}{2}e_{4}^{2}$, and taking the time-derivative of $V_{4}$ along (Equation 99) yields(100)$$\begin{array}{c} {\dot{V}}_{4}=e_{4}{\dot{e}}_{4}=-k_{4}\left|e_{4}\right|+\xi \left(t\right)e_{4}\leq -\left(k_{4}-\left|\bar{\xi }\right|\right)\left|e_{4}\right| \end{array}$$It is observed that ${\dot{V}}_{4}\leq 0$ in (Equation 100) by choosing the design parameter(101)$$k_{4}>\bar{\xi }$$
by which (Equation 100) is rewritten as(102)$${\dot{V}}_{4}\leq -\sqrt{2}{\bar{k}}_{4}{V_{4}}^{\frac{1}{2}}$$
where ${\bar{k}}_{4}≜k_{4}-\bar{\xi }$ and $\left|e_{4}\right|={\left(2V_{4}\right)}^{\frac{1}{2}}$.Therefore, by combining (Equation 102) with *Lemma 1*, we obtain that the Lyapunov function $V_{4}$ can converge from the initial value $V_{4}{\left(e_{4}\right)}_{t=T_{4}}$ to zero in finite-time $t\in \left[T_{4},\;T_{4}+t_{7}\right)$, satisfying(103)$$t_{7}\leq \frac{\sqrt{2}}{{\bar{k}}_{4}}{\left(V_{4}{\left(e_{4}\right)}_{t=T_{4}}\right)}^{\frac{1}{2}}$$In summary, the ESO estimation errors $e_{j},\;j=0,\;1,\;2,\;3,\;4$ converge to zero in finite time $t\leq T_{5}≜T_{4}+t_{7}<\infty$. □

### 3.3. Global Sliding Mode Function Construction

Since the designed ESO is finite-time convergent, satisfying the separation principle [49], the output feedback controller is designed in this part using the ESO estimations.

Define a nonlinear time function as follows:(104)$$s_{0}\left(t\right)=-\bar{s}{\left(z\right)}_{t=0}\cdot \left(A_{1}e^{-B_{1}t}+A_{2}e^{-B_{2}t}+A_{3}e^{-B_{3}t}\right)$$
where the design parameters $A_{j},\;B_{j}>0$, j=1,2,3; $\bar{s}\left(z\right)=z_{3}+\sum_{i=1}^{2}c_{i}z_{i}\;,\;c_{i}>0$, i=1,2; and the design parameters $c_{i}$ are selected such that(105)$$P^{2}+c_{2}P+c_{1}=0$$
is Hurwitz, where *P* is the Laplace operator.

Using (Equation 24) and (Equation 104), the sliding mode surface is constructed as(106)$$s\left(z,t\right)=\bar{s}\left(z\right)+s_{0}\left(t\right)$$

Notice that the introduction of $s_{0}\left(t\right)$ is to reduce the reaching mode of the sliding surface, such that the closed-loop system stays at the sliding mode motion stage from the initial moment, which can enhance the robustness of the system [56]. In order to achieve this goal, the selection of the design parameters should meet the following conditions:

1. $s_{0}\left(0\right)=-\bar{s}{\left(z\right)}_{t=0}$, i.e.,(107)$$A_{1}+A_{2}+A_{3}=1$$

2. There exists $0<t_{f}<\infty$, $\forall t\geq t_{f}$, such that $s_{0}\left(t\right)=0$ and $s_{0}\left(t_{f}\right)=0$, i.e.,(108)$$A_{1}e^{-B_{1}t_{f}}+A_{2}e^{-B_{2}t_{f}}+A_{3}e^{-B_{3}t_{f}}=0$$

Using (Equation 107) and (Equation 108), we have(109)$$\begin{array}{cc} \; & A_{3}=1-A_{1}-A_{2}\\ \; & A_{2}e^{-B_{2}t_{f}}+A_{3}e^{-B_{3}t_{f}}=-A_{1}e^{-B_{1}t_{f}} \end{array}$$
It is easily obtained by (Equation 109) that(110)$$\begin{array}{cc} \; & A_{2}=\frac{\left(1-A_{1}\right)e^{-B_{3}t_{f}}+A_{1}e^{-B_{1}t_{f}}}{e^{-B_{3}t_{f}}-e^{-B_{2}t_{f}}}\\ \; & A_{3}=\frac{-A_{1}e^{-B_{1}t_{f}}-\left(1-A_{1}\right)e^{-B_{2}t_{f}}}{e^{-B_{3}t_{f}}-e^{-B_{2}t_{f}}} \end{array}$$

Applying (Equation 110) into (Equation 104) yields(111)$$s_{0}\left(t\right)=\left\{\begin{array}{cc} \begin{array}{c} -\bar{s}{\left(z\right)}_{t=0}\cdot (A_{1}e^{-B_{1}t_{f}}\\ +\frac{\left(1-A_{1}\right)e^{-B_{3}t_{f}}+A_{1}e^{-B_{1}t_{f}}}{e^{-B_{3}t_{f}}-e^{-B_{2}t_{f}}}e^{-B_{2}t}\\ +\frac{-A_{1}e^{-B_{1}t_{f}}-\left(1-A_{1}\right)e^{-B_{2}t_{f}}}{e^{-B_{3}t_{f}}-e^{-B_{2}t_{f}}}e^{-B_{3}t}), \end{array} & t\leq t_{f}\\ 0, & t>t_{f} \end{array}\right.$$

Therefore, for a given set of design parameters $A_{1}$, $B_{1}$, $B_{2}$, and $B_{3}$, the values of $A_{2}$ and $A_{3}$ can be obtained by (Equation 110), and thus, the sliding mode surface can be constructed by (Equation 106). Specifically, the nonlinear time function defined by (Equation 111) satisfies the above global sliding mode conditions.

In addition, to ensure that $s_{0}\left(t\right)$ is globally differentiable, the following condition must hold:(112)$$\lim_{t\to t_{f}^{+}}\;\frac{d}{dt}s_{0}\left(t\right)=\lim_{t\to t_{f}^{-}}\;\frac{d}{dt}s_{0}\left(t\right)=0$$

Combining (Equation 112) with (Equation 111) yields $\forall B_{2}\neq B_{3}$,(113)$$\begin{array}{cc} \; & \left(B_{3}-B_{1}\right)A_{1}e^{-\left(B_{1}+B_{3}\right)t_{f}}+\left(B_{1}-B_{2}\right)A_{1}e^{-\left(B_{1}+B_{2}\right)t_{f}}\\ \; & +\left(1-A_{1}\right)\left(B_{3}-B_{2}\right)e^{-\left(B_{2}+B_{3}\right)t_{f}}=0 \end{array}$$

The above analysis shows that with the given design parameters $A_{1},B_{1},B_{2},B_{3},\left(B_{2}\neq B_{3}\right)$, the value of $t_{f}$ is solved by (Equation 113), and the nonlinear time function is constructed by (Equation 111). Thereafter, the sliding mode surface can be constructed by substituting the above results into (Equation 106).

### 3.4. Event-Triggered Control Input

Define the measurement error as $\Delta =\varpi \left(t\right)-\varpi \left(t_{\dot{k}}\right)$, the event-triggering mechanism is designed as(114)$$t_{k+1}=\inf\left\{t\in R\left|\begin{array}{c} \left|\Delta \right|\geq C,\left|u\left(t\right)\right|\geq \Gamma \\ \left|\Delta \right|\geq \delta \left|\varpi \left(t_{k}\right)\right|+D,\left|u\left(t\right)\right|<\Gamma \end{array}\right.\right\}$$
where k∈Z is the subscript that records the triggering time, $t_{0}=0$ is the initial time, the design parameters *C*, *D*, Γ>0, and 0<δ<1.

The piecewise continuous event-triggered controller is designed as(115)$$u\left(t\right)=-\frac{1}{c_{1}+c_{2}{\bar{b}}_{2}+{\bar{b}}_{3}}\cdot \varpi \left(t_{k}\right)$$
where the continuous signal(116)$$\begin{array}{cc} \varpi \left(t\right)= & {\bar{c}}_{1}v_{1}+{\bar{c}}_{2}v_{2}+{\bar{c}}_{3}v_{3}+v_{4}+{\dot{s}}_{0}+k_{vs}v_{s}+k_{s}{\left|v_{s}\right|}^{\alpha_{s}}\mathrm{sign}\left(v_{s}\right) \end{array}$$
and $v_{i}$, i=1,2,3,4 are the estimations of $z_{i}$ obtained by the designed ESO (Equation 25), $v_{s}=v_{3}+\sum_{i=1}^{2}c_{i}v_{2}+s_{0}\left(t\right)$; ${\bar{b}}_{2}$ and ${\bar{b}}_{3}$ are defined in (Equation 22); the design parameters ${\bar{c}}_{1}=c_{1}a_{11}$, ${\bar{c}}_{2}=c_{1}+c_{2}a_{22}+{\bar{a}}_{32}$, ${\bar{c}}_{3}=c_{2}+{\bar{a}}_{33}$, $k_{vs}>0$, $k_{s}>0$, and $0<\alpha_{s}<1$.

### 3.5. Stability Analysis

The stability analysis of the proposed IGC design is summarized in *Theorem 3*.

**Theorem** **3.**
*Considering the AUV IGC model (Equation 14) satisfying Assumptions 1–3, the output feedback event-triggered sliding mode controller given by (Equation 25), (Equation 114), (Equation 115) and (Equation 116) guarantees the finite-time stability of the closed-loop system; the state error can converge to an arbitrarily small adjustable neighborhood containing the origin in finite time.*


**Proof.** Define the Lyapunov function(117)$$V_{s}=\frac{1}{2}s^{2}$$Taking the time-derivative of (Equation 117) along (Equation 22) and applying (Equation 115), we obtain(118)$$\begin{array}{cc} {\dot{V}}_{s}= & s(c_{1}a_{11}z_{1}+\left(c_{1}+c_{2}a_{22}+{\bar{a}}_{32}\right)z_{2}+z_{4}\;\;\;+\left(c_{2}+{\bar{a}}_{33}\right)z_{3}+\left(c_{1}+c_{2}{\bar{b}}_{2}+{\bar{b}}_{3}\right)u+\dot{s})\\ = & s\left({\bar{c}}_{1}z_{1}+{\bar{c}}_{2}z_{2}+{\bar{c}}_{3}z_{3}+z_{4}\right.\;\;\;\left.-\varpi \left(t_{k}\right)+{\dot{s}}_{0}-\varpi \left(t\right)+\varpi \left(t\right)\right) \end{array}$$Substituting (Equation 116) into (Equation 118) yields(119)$$\begin{array}{cc} {\dot{V}}_{s}= & s\left({\bar{c}}_{1}e_{1}+{\bar{c}}_{2}e_{2}+{\bar{c}}_{3}e_{3}+e_{4}\right)-k_{vs}sv_{s}\;\;\;-sk_{s}{\left|v_{s}\right|}^{\alpha_{s}}\mathrm{sign}\left(v_{s}\right)-s\left(\varpi \left(t_{k}\right)-\varpi \left(t\right)\right) \end{array}$$Notice that the measurement error is $\Delta =\varpi \left(t\right)-\varpi \left(t_{k}\right)$; applying *Lemma 4*, the right side of (Equation 119) satisfies(120)$$\begin{array}{cc} s\left({\bar{c}}_{1}e_{1}+{\bar{c}}_{2}e_{2}+{\bar{c}}_{3}e_{3}+e_{4}\right) & \leq \;\;\;\frac{{\bar{c}}_{1}e_{1}^{2}+{\bar{c}}_{2}e_{2}^{2}+{\bar{c}}_{3}e_{3}^{2}+e_{4}^{2}}{2}+\frac{{\bar{c}}_{1}+{\bar{c}}_{2}+{\bar{c}}_{3}+1}{2}s^{2}-k_{vs}sv_{s}\\ \; & =-k_{vs}s\left(s-e_{s}\right)\leq -\frac{k_{vs}}{2}s^{2}+\frac{k_{vs}}{2}e_{s}^{2}\\ \; & \;\;\;\;\;\;\;s\Delta \leq \frac{1}{2}s^{2}+\frac{1}{2}\Delta^{2} \end{array}$$
where $e_{s}=s-v_{s}$.Applying *Lemma 5*, we obtain(121)$$\begin{array}{cc} -sk_{s}{\left|v_{s}\right|}^{\alpha_{s}}sign\left(v_{s}\right) & =-sk_{s}{\left|s-e_{s}\right|}^{\alpha_{s}}sign\left(s-e_{s}\right)\leq -k_{s}{\left|s\right|}^{\left(\alpha_{s}+1\right)}+\frac{k_{s}}{2}s^{2}+2k_{s}e_{s}^{2\alpha_{s}} \end{array}$$Combining (Equation 120) and (Equation 121) with (Equation 119) yields(122)$$\begin{array}{cc} {\dot{V}}_{s}\leq & -\frac{1}{2}\left(k_{vs}-{\bar{c}}_{1}-{\bar{c}}_{2}-{\bar{c}}_{3}-2-k_{s}\right)s^{2}-k_{s}{\left|s\right|}^{\left(\alpha_{s}+1\right)}\\ & +\frac{{\bar{c}}_{1}}{2}e_{1}^{2}+\frac{{\bar{c}}_{2}}{2}e_{2}^{2}+\frac{{\bar{c}}_{3}}{2}e_{3}^{2}+\frac{1}{2}e_{4}^{2}+\frac{k_{vs}}{2}e_{s}^{2}+2k_{s}e_{s}^{2\alpha_{s}}+\frac{1}{2}\Delta^{2} \end{array}$$Select the appropriate design parameter $k_{vs}>0,k_{s}>0$ such that ${\bar{k}}_{s}≜k_{\mathrm{vs}}-{\bar{c}}_{1}-{\bar{c}}_{2}-{\bar{c}}_{3}-2-k_{s}>0;$ and define$$E≜\frac{{\bar{c}}_{1}}{2}e_{1}^{2}+\frac{{\bar{c}}_{2}}{2}e_{2}^{2}+\frac{{\bar{c}}_{3}}{2}e_{3}^{2}+\frac{1}{2}e_{4}^{2}+\frac{k_{vs}}{2}e_{s}^{2}+2k_{s}e_{s}^{2\alpha_{s}}$$
as the error caused by the ESO estimation; then, (Equation 122) can be rewritten as(123)$$\begin{array}{c} {\dot{V}}_{s}\leq -\frac{1}{2}{\bar{k}}_{s}s^{2}-k_{s}{\left|s\right|}^{\left(\alpha_{s}+1\right)}+E+\frac{1}{2}\Delta^{2}=-{\bar{k}}_{s}V_{s}-2^{\frac{\left(\alpha_{s}+1\right)}{2}}k_{s}V_{s}^{\frac{\left(\alpha_{s}+1\right)}{2}}+E+\frac{1}{2}\Delta^{2} \end{array}$$
where $s=\sqrt{2V_{s}}$ using (Equation 117).According to *Theorems 1 and 2*, the error *E* is globally bounded and converges to zero in finite time. In addition, according to (Equation 114) and (Equation 115), the measurement error satisfies$$\Delta \leq \bar{\Delta }≜\max\left\{C,\delta |c_{1}+c_{2}{\bar{b}}_{2}+{\bar{b}}_{3}|\Gamma +D\right\}$$
which means that $E+\frac{1}{2}\Delta^{2}$ is globally bounded.Combining (Equation 123) with *Lemma 2* gives(124)$${\dot{V}}_{s}+{\bar{k}}_{s}V_{s}+2^{\frac{\alpha_{s}+1}{2}}k_{s}V_{s}^{\frac{\alpha_{s}+1}{2}}\leq \bar{E}≜E_{t=0}+\frac{1}{2}{\bar{\Delta }}^{2}$$Therefore, the closed-loop system is finite-time stable and the errors converge to the set(125)$$\begin{array}{cc} \; & U_{s}=\left\{\left.s\right|V_{s}\leq \min\left\{\frac{\bar{E}}{\left(1-c_{0}\right){\bar{k}}_{s}},\;f\left(\bar{E}\right)\right\}\right\} \end{array}$$
where $f\left(\bar{E}\right)={\left(\frac{\bar{E}}{\left(1-c_{0}\right)\cdot 2^{\frac{\alpha_{s}+1}{2}}\cdot {\bar{k}}_{s}}\right)}^{\frac{\alpha_{s}+1}{2}}$, $o<c_{0}<1$.The convergence time satisfies(126)$$\begin{array}{cc} t & \leq \max\left\{\ln{\left(\frac{c_{0}{\bar{k}}_{s}\cdot {\left(V{\left(x\right)}_{t=0}\right)}^{\left(\frac{1-\alpha_{s}}{2}\right)}+2^{\frac{\alpha_{s}+1}{2}}\cdot k_{s}}{2^{\frac{\alpha_{s}+1}{2}}\cdot k_{s}}\right)}^{\frac{1}{c_{0}}},\right.\\ \; & \left.\ln\frac{{\bar{k}}_{s}\cdot {\left(V{\left(x\right)}_{t=0}\right)}^{\left(\frac{1-\alpha_{s}}{2}\right)}+c_{0}\cdot 2^{\frac{\alpha_{s}+1}{2}}\cdot k_{s}}{c_{0}\cdot 2^{\frac{\alpha_{s}+1}{2}}\cdot k_{s}}\right\}\cdot \frac{2}{{\bar{k}}_{s}\left(1-\alpha_{s}\right)} \end{array}$$Combining (Equation 125) with (Equation 105), (Equation 106) and (Equation 111), it can be seen that the system state $z_{1}≜r\dot{q}$ eventually converges to a non-zero residual set containing the origin, which can be made arbitrarily small to satisfy the required control performance by adjusting the design parameters $k_{vs}$ and $k_{s}$.Notice that r>0. It is proven that $\left|\dot{q}\right|$ converges to an adjustable neighborhood about the origin in finite time, i.e., the AUV can intercept the target in finite time. Since all signals are globally bounded, by Theorems 1 and 2 of [57], the inter-event interval of the event-triggering mechanism (Equation 114) is lower bounded by positive constants excluding the Zeno phenomenon. □

## 4. Simulations

Considering the AUV lateral dynamics in [39], the initial states are given by (x(0),y(0))=(0,10) m, $\psi_{A}\left(0\right)=\sigma_{A}\left(0\right)=-45{}^{\circ},\;\beta \left(0\right)=0{}^{\circ},\;\omega_{y}\left(0\right)=0\;\mathrm{rad}/s$, and $v_{A}=25\;m/s$; the initial position of the incoming target is (1000,−800) m, $\sigma_{T}\left(0\right)=136.05{}^{\circ}$, $v_{T}=25\;m/s$; the unmeasurable target maneuvering angular velocity is $\omega_{T}=2\cos(1+0.1t){}^{\circ}/s$; and the *x* and *y* coordinates of the target are measured with random errors satisfying the uniform distribution of (0,3) m.

The design parameters of the ESO are given by $\rho_{i}=2$, $\alpha_{0}=0.85$, $\alpha_{i}=\left(i+1\right)\alpha_{0}-i$, $\beta_{0}=1/\alpha_{0}$, $\beta_{i}=\beta_{0}+i\left(\alpha_{0}-1\right)$, and i=0,1,2,3,4; $k_{0}=10,\;k_{1}=40$, $k_{2}=80$, $k_{3}=80$, and $k_{4}=32$. The controller design parameters are given by $c_{1}=12$, $c_{2}=2$, $A_{1}=0.5$, $B_{1}=150$, $B_{2}=100$, $B_{3}=10$, $t_{f}=0.011$, $k_{vs}=5$, $k_{s}=1$, and $\alpha_{s}=0.7$; the control rudder is constrained as $\left|\delta_{r}\right|\leq 15{}^{\circ}$; and the event-triggering mechanism design parameters are given by Γ=10, C=1, D=0.05, and δ=0.1.

In order to verify the effectiveness of the proposed method, simulation comparisons were carried out with the traditional proportional guidance, trail guidance, and fixed advanced angle guidance (the advanced angle is 10°). The simulation results are shown in Figure 3, Figure 4, Figure 5, Figure 6 and Figure 7. It can be seen from Figure 3 and Figure 4 that the states of the AUV can be globally bounded by the proposed IGC method. Figure 5a illustrates the trajectories of the AUV and target. Combining with Figure 5b and Table 1, it is obtained that the proposed IGC method can intercept the target with a smaller miss distance compared with the traditional guidance and control methods. Figure 6 shows the curve of the control rudder input that is updated at the triggering time; and the proposed event-triggered controller has all triggering intervals greater than zero, which avoids the occurrence of the Zeno phenomenon. Figure 7 depicts the estimation errors of the designed finite-time ESO, which demonstrates that the estimation errors converges rapidly within a short time frame, indicating the observer’s capability to carry out fast and accurate estimation of the unmeasurable signals.

## 5. Conclusions

This paper presents an IGC framework for AUVs intercepting maneuvering targets in complex marine environments with unknown disturbances and unmeasurable states. By leveraging a state transformation method, the proposed scheme converts the original system with unmatched disturbances into a form amenable to effective observer and controller design. A finite-time ESO is developed to estimate the unknown states and disturbances. An event-triggered sliding mode controller is then designed using these estimates, achieving finite-time stability of the closed-loop system while reducing communication and actuator burdens. Simulation results validate the superiority of the proposed method in enhancing interception accuracy and responsiveness compared to traditional approaches, while effectively addressing the challenges of unknown disturbances and unmeasurable states. The designed scheme demonstrates significant potential for practical implementation in AUV interception tasks.

Future work will focus on extending the proposed framework to three-dimensional interception scenarios and incorporating adaptive mechanisms to handle time-varying hydrodynamic parameters. Additionally, experimental validation will be conducted to further verify the practicality of the proposed method.

## Figures and Tables

**Figure 1 sensors-25-03088-f001:**
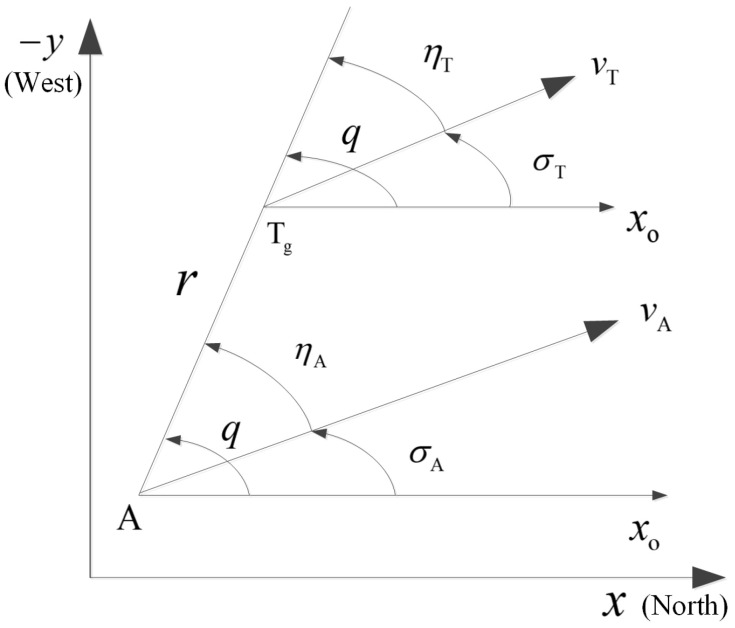
Interception geometry between the AUV and target.

**Figure 2 sensors-25-03088-f002:**
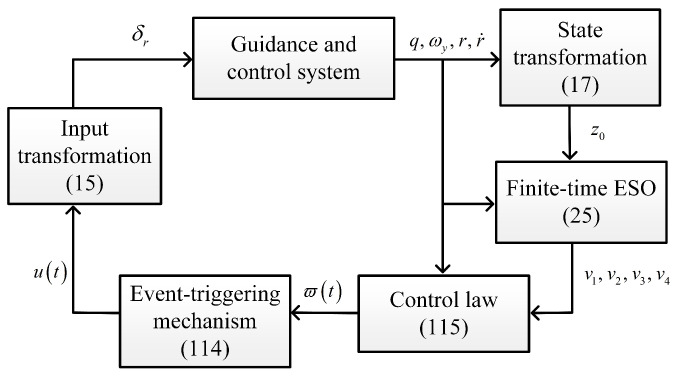
IGC design diagram.

**Figure 3 sensors-25-03088-f003:**
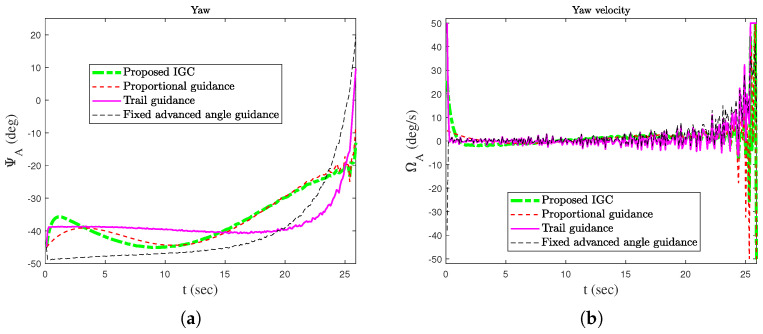
Attitude cures of the AUV. (**a**) Yaw. (**b**) Yaw velocity.

**Figure 4 sensors-25-03088-f004:**
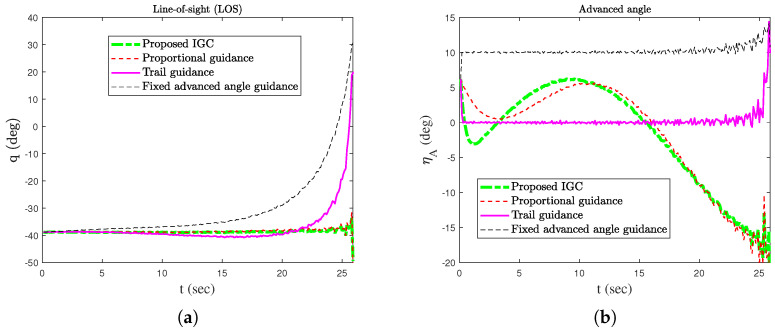
LOS and advanced angle. (**a**) LOS. (**b**) Advanced angle.

**Figure 5 sensors-25-03088-f005:**
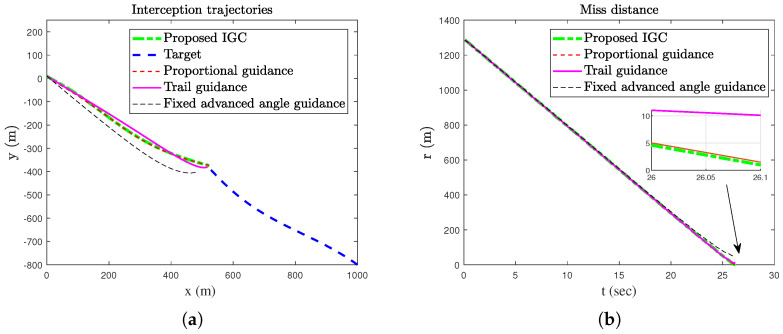
The trajectories and miss distance of the AUV and the target. (**a**) Trajectories of the AUV and target. (**b**) Miss distance.

**Figure 6 sensors-25-03088-f006:**
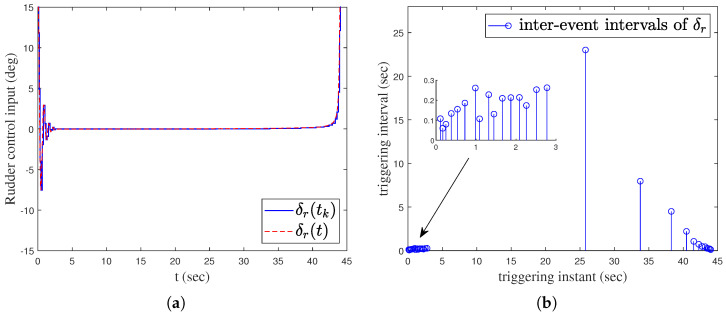
Event-triggered sliding mode controller. (**a**) Rudder control input. (**b**) Inter-event intervals.

**Figure 7 sensors-25-03088-f007:**
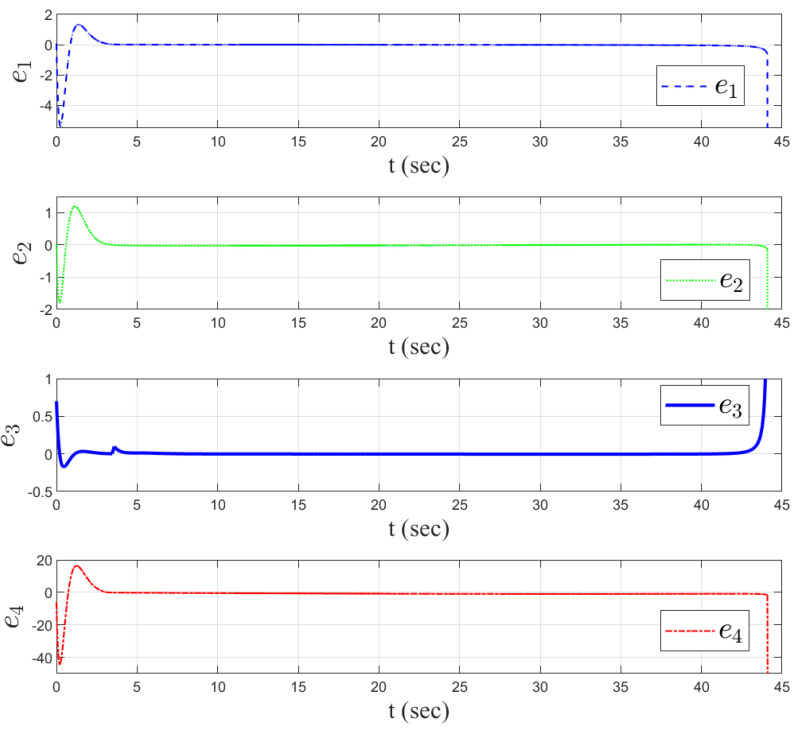
Estimate errors of the finite-time ESO.

**Table 1 sensors-25-03088-t001:** Comparison of miss distance.

Algorithm	Miss Distance (m)
Proposed IGC	1.86
Proportional guidance	2.81
Trail guidance	12.97
Fixed advanced angle guidance	49.67

## Data Availability

The data presented in this study are available on request from the corresponding author due to confidentiality regulations.

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
