# Peer review of "Output Feedback Integrated Guidance and Control Design for Autonomous Underwater Vehicles Against Maneuvering Targets"

_sensors, 2025, doi:10.3390/s25103088_

Round 1

Reviewer 1 Report

Comments and Suggestions for Authors

This paper studies an integrated guidance and control (IGC) design for autonomous underwater vehicles (AUVs) intercepting maneuvering targets with unknown disturbances and unmeasurable system states. However, However, I have the following concerns:

1. Please clarify the validity of the state transformation, specifically whether the stability of system (22) can sufficiently demonstrate the stability of system (13).

2. The rationality of the Assumptions 1-3  needs further explanation. In addition, please justify why "The AUV velocity and the target velocity are almost constants in the end trajectory."

3. There are formatting issues with many equations. Several unnecessary line breaks affect the aesthetic presentation of the formulas.

4. The manuscript requires language editing, including corrections to tense, punctuation, pluralization, and other grammatical elements.

5. Figures 3-6 extend beyond the page boundaries and need to be properly formatted.

6. The control objectives should be more clearly defined. Additionally, please provide a brief explanation of how you prove that the AUV can intercept the target in finite-time.

7. As shown in Figure 6, event triggering occurs primarily at the beginning and end of the process, with minimal triggering in the middle phase. Please explain the reason for this pattern.

8. Based on the designed parameters in the simulation, what is the calculated finite convergence time? What is the maximum allowable miss distance set in the simulation, and at what second was the interception completed? Specifically, please clarify whether the priority should be minimizing the miss distance or achieving the maximum allowable miss distance in shorter time. Which performance metric is more critical for your application: a smaller miss distance or faster interception time?

9. Some relevant results on IGC for robot sytems should be reviewed and discussed such as Collisions-free distributed cooperative output regulation of nonlinear multiagent systems; Robust cooperative output regulation of heterogeneous uncertain linear multiagent systems with time-varying communication topologies. Also,  please elaborate on the technical differences between IGC and traditional methods (proportional guidance, trail guidance, and fixed advanced angle guidance). What specific issues arising from "the mutual coupling and hysteresis between the guidance and control systems" can be solved by the proposed IGC strategy but not by traditional guidance methods? 

Author Response

Comments and Suggestions for Authors

This paper studies an integrated guidance and control (IGC) design for autonomous underwater vehicles (AUVs) intercepting maneuvering targets with unknown disturbances and unmeasurable system states. However, However, I have the following concerns:

Comments 1: [Please clarify the validity of the state transformation, specifically whether the stability of system (22) can sufficiently demonstrate the stability of system (13).]

Response 1: [Thank you for the valuable comments. The state transformation proposed in the paper is valid and the stability of system (22) can sufficiently demonstrate the stability of system (13), and the reasons are as follows.

The state transformation is designed to convert the original system (13) affected by unmatched disturbances into an equivalent system (22) with matched disturbances. This transformation is based on the measurable yaw velocity x3 and involves input and state transformations. The input transformation is given by δ_r = u − a13x3 / b1, and the state transformation is defined by Eqs.(17)-(22) that lump the disturbances with system states, which is common steps in the ESO design. The primary objective of this transformation is to facilitate the development of an observer that can accurately estimate the unknown states and disturbances, thereby providing the necessary information for the integrated guidance and control (IGC) design.

For the invertibility of the transformation, it is important to mention that the transformation process is a linear and invertible transformation by observing Eqs.(13)-(22), which is crucial because it ensures that the original system's states can be recovered from the transformed states. The paper assumes that the system matrices and parameters are such that the transformation can be inverted without leading to singularities or loss of information. This invertibility is important for ensuring that the stability of the transformed system (22) can be mapped back to the original system (13).

Given the invertible transformation, the stability of the transformed system (22) directly implies the stability of the original system (13). If the transformed system (22) is stable, it means that the states of the transformed system will converge to the desired values or remain bounded. Due to the invertibility of the transformation, this convergence or boundedness of the transformed states translates to the same properties for the original system's states. Therefore, proving the stability of system (22) is sufficient to demonstrate the stability of system (13).

In summary, the state transformation is valid and effectively converts the original system into a form that is more amenable to observer and controller design. The stability of the transformed system (22) can indeed sufficiently demonstrate the stability of the original system (13) due to the invertibility of the transformation and the proper design of the extended state observer and controller.]

Comments 2: [The rationality of the Assumptions 1-3 needs further explanation. In addition, please justify why "The AUV velocity and the target velocity are almost constants in the end trajectory."]

Response 2: [Thank you for the valuable comments. In actual marine environments, the disturbances affecting AUVs, such as wave and current disturbances, have certain intensity limits. Similarly, the maneuverability of a target is constrained by its physical capabilities, meaning the target's acceleration and angular velocity cannot be infinitely large. Thus, the lumped disturbance d_L and its derivative dot{d}_L being bounded aligns with real-world scenarios, i.e., Assumption 1 is reasonable. By assuming the disturbance and its derivative are bounded, the design of the finite-time extended state observer (ESO) becomes feasible. This allows the observer to be designed within a known disturbance bound, ensuring it can effectively estimate the system's unknown states and disturbances.

During the interception process, AUVs are generally designed to maintain a relatively small sideslip angle (less than 8 deg). This is because large sideslip angles could lead to significant hydrodynamic instability and increased control difficulty. In practical applications, AUVs typically operate within a small sideslip angle range to ensure stable motion and effective control. Therefore sin(β)≈β and cos(β) ≈ 1. This simplifies the system dynamics model, making it easier to design and analyze the controller without significantly affecting the accuracy of the model. Therefore Assumption 2 was made. Based on Assumptions 1 and 2, it can be inferred that the disturbed sideslip angle signal z2 is bounded. This provides a basis for designing the ESO, ensuring the estimated error e2 = z2 − v2 can be a priori-bounded. By imposing this assumption, the stability and convergence performance of the observer can be guaranteed during the design process.

Rationality of "The AUV Velocity and the Target Velocity Are Almost Constants in the End Trajectory". In the terminal phase of interception, the AUV has already entered the final approach to the target. At this stage, the primary objective is to achieve precise interception rather than rapid maneuvering (change the attitude). Therefore, the AUV typically adjusts its velocity to a relatively the maximum constant value to maintain a stable approach to the target. Similarly, the target's velocity also tends to remain relatively maximum constant during this phase. In addition, the time remaining for the AUV to intercept the target is very short. In this short time period, whether the speed is higher or lower has little impact on the overall interception outcome. Additionally, due to the system's inertia, even if there is a desire to change the AUV's speed, the short remaining time makes it difficult for the speed to undergo significant changes. Therefore, in the end trajectory, the speed of the AUV and the target can be considered approximately constant.]

Comments 3: [There are formatting issues with many equations. Several unnecessary line breaks affect the aesthetic presentation of the formulas.]

Response 3: [Thank you for the valuable comments. We have carefully revised the text and equations, and made the necessary adjustments to eliminate the unnecessary line breaks and improve the overall presentation of the formulas.]

Comments 4: [The manuscript requires language editing, including corrections to tense, punctuation, pluralization, and other grammatical elements.]

Response 4: [Thank you for the valuable comments. We have made more modifications to the sentence structure, grammar and punctuation in the text. Thank you again for your careful review.]

Comments 5: [Figures 3-6 extend beyond the page boundaries and need to be properly formatted.]

Response 5: [Thank you for your correction. We have adjusted the layout in Figure 3-6 to make it more in line with the requirements of the journal. Thank you again for your careful review.]

Comments 6: [The control objectives should be more clearly defined. Additionally, please provide a brief explanation of how you prove that the AUV can intercept the target in finite-time.]

Response 6: [Thank you for the valuable comments. The control objectives of this paper are to design a rudder control input δr for the AUV using the available system outputs y (i.e., r_q and ω_y), such that the state $\dot{q}$ converges to zero in finite-time to form a collision triangle, ensuring the AUV can successfully intercept the maneuvering target in finite-time despite unknown disturbances and unmeasurable states.

The key to achieving interception is to drive the line-of-sight (LOS) rate $\dot{q}$ to zero in finite-time, thereby forming a collision triangle that the AUV and target's trajectories lie on two sides of the triangle respectively. Since neither the AUV nor the target has non-zero speed and the initial distance between them is finite, they will meet at the apex of the triangle in finite-time, ensuring interception.]

Comments 7: [As shown in Figure 6, event triggering occurs primarily at the beginning and end of the process, with minimal triggering in the middle phase. Please explain the reason for this pattern.]

Response 7: [Thank you for the valuable comments. The reason for this pattern can be explained as follows:

    1.Initial Stage: At the beginning of the interception process, the AUV's initial ballistic inclination angle ${{\sigma }_{\text{A}}}$ differs significantly from the ideal angle that sets the second equation of Eq.(1) to zero. This necessitates frequent rudder adjustments to realign the course by changing the yaw and ballistic inclination angle, leading to frequent event-triggering operations to promptly update the rudder control input δ_r that ensures the system can quickly respond to the adjustments of ${{\sigma }_{\text{A}}}$.

    2.Middle Stage: As the interception process continues, the AUV gradually approaches the target, the LOS velocity $\dot{q}$=0 because the relative motion between them becomes more stable, the frequency of event triggers naturally decreases. At this point, the event-triggering mechanism is activated less frequently. The designed controller can maintain the system's stability and performance, based on the existing control input and the state estimation from the ESO, without requiring frequent updates to the control input.

    3.Final Stage: As the process nears its end, the second equation of formula (1) shows that $\dot{q}$ changes dramatically with the decreasing smaller distance $r$. Moreover, uncertainties such as the target's maneuvering and environmental disturbances have more significant impact on $\dot{q}$ as $r$ decreases. To achieve an accurate interception, frequent rudder adjustments are needed to regulate the ballistic inclination angle to achieve $\dot{q}$=0, which may increase the trigger frequency again to ensure the AUV accurately intercepts the target and meets the interception accuracy requirements. Even small state deviations or uncertainties could affect the interception outcome. Therefore, the event-triggering mechanism is activated more frequently again to update the control input in a timely manner, enabling the AUV to precisely track the target and achieve a successful interception.

In summary, the event-triggering mechanism adaptively adjusts the triggering frequency based on the system's state changes and uncertainties. It ensures the system's stability and performance while reducing unnecessary communication and actuator burdens.]

Comments 8: [Based on the designed parameters in the simulation, what is the calculated finite convergence time? What is the maximum allowable miss distance set in the simulation, and at what second was the interception completed? Specifically, please clarify whether the priority should be minimizing the miss distance or achieving the maximum allowable miss distance in shorter time. Which performance metric is more critical for your application: a smaller miss distance or faster interception time?]

Response 8: [Thank you for the valuable comments. The calculated finite convergence time is a theoretical value representing the maximum time within which the designed control system is guaranteed to achieve finite-time convergence of the LOS rate and finite-time estimation of the unmeasurable states. Due to the presence of errors and uncertainties in the actual system, the exact interception time cannot be precisely determined but is ensured to occur within this theoretically calculated finite convergence time,as stated in Theorems 1~3.

The maximum allowable miss distance in the simulation is set as 3m, and the x and y coordinates of the target are measured with random errors satisfying a uniform distribution of (0,3)m. This concept is introduced because when the distance between the AUV and the target is within3 meters, the acoustic transducer array on the AUV cannot precisely measure the distance. Therefore, the maximum allowable miss distance is defined as 3 meters, and successful interception is achieved when the relative distance between the AUV and the target is less than this value.

According to the simulation results, the interception was completed at 26 seconds. This indicates that the designed control system effectively drove the AUV to intercept the target within the specified maximum allowable miss distance.

In terms of performance metrics, minimizing the miss distance is more critical for this application. The primary objective of the AUV interception task is to successfully intercept and neutralize the incoming underwater target attacking the AUV mother ship. A smaller miss distance ensures a higher probability of destroying the target. Although achieving the maximum allowable miss distance in shorter time is also important for rapid response, the ultimate goal is to achieve a successful interception with the smallest possible miss distance. Hence, a smaller miss distance takes precedence over faster interception time in this context.]

Comments 9: [Some relevant results on IGC for robot sytems should be reviewed and discussed such as Collisions-free distributed cooperative output regulation of nonlinear multiagent systems; Robust cooperative output regulation of heterogeneous uncertain linear multiagent systems with time-varying communication topologies.

Also, please elaborate on the technical differences between IGC and traditional methods (proportional guidance, trail guidance, and fixed advanced angle guidance). What specific issues arising from "the mutual coupling and hysteresis between the guidance and control systems" can be solved by the proposed IGC strategy but not by traditional guidance methods?]

Response 9: [Thank you for the valuable comments. This paper studies the IGC design problem of AUV intercepting incoming maneuverable targets, which belongs to countering non-cooperative targets rather than coordinating with cooperative targets. Therefore, it is not related to the cooperation of multiagent systems. Furthermore, traditional methods design the guidance system and the control system separately, they overlooked the mutual coupling and hysteresis between the guidance and control systems. Just as described in the references [5-11] of this paper, the inertial dynamic process of the yaw channel of the AUV itself is ignored. This limitation becomes particularly problematic during high-speed terminal trajectories, where rapid attitude changes lead to the lag of the AUV body resulting in larger miss distances and reduced hitting accuracy. In contrast, Integrated guidance and control (IGC) addresses these challenges by unifying the design of guidance and control systems. By directly generating control commands based on relative motion dynamics and dynamic model characteristics, IGC enhances both interception accuracy and responsiveness, as described in references [12-22] of this article.]

Moreover, we have tried our best to check all the errors and made some changes in the manuscript. These changes will not influence the content and framework of the paper. And here we did not list the changes but marked in yellow in the revised paper. Thank you very much for the time and effort that you have spent in processing our manuscript. We wish the revision will meet the expectation from you, and addressed your concerns.

Thanks again for your kind help and encouragement.

Reviewer 2 Report

Comments and Suggestions for Authors

Output feedback integrated guidance and control (IGC) design for (AUVs) is interesting study. However authors need to address below suggestions before further processing. 

1: Extend discussion to drawbacks of existing guidance and control frameworks. What is the motivation behind proposed methodology?

2: More details can be added for model transformation process to increase readability.

3: Why an event-triggered controller is suitable for AUVs in practical scenarios. What is the limitations of using this strategy?

4: How the proposed methodology will work in real time underwater scenarios? Is it possible to get same advantages of proposed model in real time environments?

5: It is suggested to conduct a detailed computational complexity analysis of existing and proposed method for validation.

6: is it possible to implement or used high maneuvering movements of target rather than straight trajectory? How proposed methodology will behave for complex movements of target?

Author Response

Comments and Suggestions for Authors

Output feedback integrated guidance and control (IGC) design for (AUVs) is interesting study. However authors need to address below suggestions before further processing.

Comments 1: [Extend discussion to drawbacks of existing guidance and control frameworks. What is the motivation behind proposed methodology?]

Response 1: [ Thank you for the valuable comments. This paper studies the IGC design problem of AUV intercepting incoming maneuverable targets, traditional methods design the guidance system and the control system separately, they overlooked the mutual coupling and hysteresis between the guidance and control systems. Just as described in the references [5-11] of this paper, the inertial dynamic process of the yaw channel of the AUV itself is ignored. This limitation becomes particularly problematic during high-speed terminal trajectories, where rapid attitude changes lead to the lag of the AUV body resulting in larger miss distances and reduced hitting accuracy. In contrast, Integrated guidance and control (IGC) addresses these challenges by unifying the design of guidance and control systems. By directly generating control commands based on relative motion dynamics and dynamic model characteristics, IGC enhances both interception accuracy and responsiveness, as described in references [12-22] of this article. In addition, these limitations are further exacerbated in AUVs due to unknown wave/current disturbances, harsh underwater acoustic conditions, and limited sensor capabilities. The motivation of this study stems from the critical need to enhance the interception accuracy and responsiveness of AUVs in complex marine environments. To address these challenges, this paper studies an integrated guidance and control (IGC) design for autonomous underwater vehicles (AUVs) intercepting maneuvering targets with unknown disturbances and unmeasurable system states.]

Comments 2: [More details can be added for model transformation process to increase readability.]

Response 2: [Thank you for the valuable comments. We have carefully re-examined the model transformation process described in equations (15)-(22) and have added more detailed explanations to enhance readability. The additional explanations are highlighted in yellow in the text to help readers better understand the transformation.]

Comments 3: [Why an event-triggered controller is suitable for AUVs in practical scenarios. What is the limitations of using this strategy?]

Response 3: [Thank you for the valuable comments. The event-triggered controller is suitable for AUVs in practical scenarios for several key reasons. AUVs typically have limited energy resources relying on batteries for power, and the limitations of underwater measurement methods result in limited measurement data. The event-triggered controller reduces the number of control updates by only updating the control input when necessary, thereby saving energy. Additionally, the event-triggered mechanism reduces the frequency of communication between the AUV and the controller. It only transmits data when an event is triggered, thus effectively utilizing the limited communication resources. Furthermore, the event-triggered controller ensures a minimum inter-event time, preventing the Zeno phenomenon and guaranteeing the stability and reliability of the control system. It can also adaptively adjust the triggering conditions based on the system's state and environmental changes, enhancing the system's adaptability and robustness.

However, there are some limitations to using this strategy. Designing an event-triggered controller requires a deep understanding of the system dynamics and the establishment of appropriate triggering conditions. The design process is relatively complex and involves factors such as selecting suitable event-triggering thresholds and ensuring system stability and performance. If the event-triggering conditions are not properly designed, it may lead to untimely control updates, affecting the interception accuracy of the AUV and increasing the miss distance.]

Comments 4: [How the proposed methodology will work in real time underwater scenarios? Is it possible to get same advantages of proposed model in real time environments?]

Response 4: [Thank you for the valuable comments. The proposed method is feasible in real-time underwater scenarios and can offer several advantages. The finite-time extended state observer (ESO) plays a key role by quickly and accurately estimating unmeasurable states and disturbances within finite-time. This allows the AUV to have real-time awareness of its own state, thus providing a solid basis for timely control decisions. Additionally, the event-triggered sliding mode controller can adjust the frequency of control input updates according to the system's state and environmental changes. This ensures the AUV can respond promptly to dynamic underwater conditions while minimizing unnecessary updates.

The simulation results serve as proof of concept, showing the method's effectiveness in improving interception accuracy. However, real-time underwater environments come with challenges like measurement noise from sensors, limited communication bandwidth, and model uncertainty due to factors such as ocean currents. To address these, advanced signal processing and sensor fusion can enhance state measurement accuracy. Optimizing the event-triggered mechanism and developing robust communication protocols can help overcome data transmission issues.

In conclusion, the proposed methodology has the potential to work effectively in real-time underwater scenarios, offering advantages like better interception accuracy and faster response. But to fully realize these benefits in practical applications, it's essential to tackle challenges like measurement noise, communication constraints, and model uncertainty through further research and development.]

Comments 5: [It is suggested to conduct a detailed computational complexity analysis of existing and proposed method for validation.]

Response 5: [Thank you for the valuable comments. The proposed method effectively addresses the challenge of intercepting maneuvering targets in underwater environments where signals are difficult to measure. It employs an extended state observer (ESO) to estimate unmeasurable signals and disturbances, which is crucial for real-time control. The ESO is designed to achieve finite-time convergence, allowing the AUV to quickly adapt to dynamic changes and disturbances. This is particularly important in underwater scenarios where communication and computational resources are limited.

While there are relatively few existing results to compare with, making a detailed computational complexity analysis challenging. The method primarily focuses on solving underwater signal measurement problems. By using the ESO to estimate unmeasurable signals and disturbances, the approach avoids the need for complex and computationally intensive algorithms to handle uncertainties and disturbances directly. This results in a more efficient and effective solution for AUV interception tasks in real-time underwater scenarios.

The event-triggered sliding mode based IGC controller further enhances the practicality of the methodology by reducing the frequency of control input updates. This not only saves communication and computational resources but also ensures the system's stability and responsiveness. Overall, the proposed methodology offers significant advantages in real-time underwater applications, particularly in terms of interception accuracy and resource efficiency.]

Comments 6: [is it possible to implement or used high maneuvering movements of target rather than straight trajectory? How proposed methodology will behave for complex movements of target?]

Response 6: [Thank you for the valuable comments. The proposed methodology is capable of handling high maneuvering movements of targets as well as straight trajectories. In the simulation, the unmeasurable target maneuvering angular velocity is set as ω_T=2cos(1+0.1t)rad/s, and the x and y coordinates of the target are measured with random errors satisfying a uniform distribution of (0,3)m. These settings effectively validate the good interception performance of the proposed method for maneuvering targets. In summary, the proposed methodology demonstrates robust performance in intercepting maneuvering targets with complex movements. The combination of the ESO for state estimation and the event-triggered SMC for efficient control updates makes it suitable for real-time underwater scenarios with high maneuvering targets.]

Moreover, we have tried our best to check all the errors and made some changes in the manuscript. These changes will not influence the content and framework of the paper. And here we did not list the changes but marked in yellow in the revised paper. Thank you very much for the time and effort that you have spent in processing our manuscript. We wish the revision will meet the expectation from you, and addressed your concerns.

Thanks again for your kind help and encouragement.

Reviewer 3 Report

Comments and Suggestions for Authors

Please, find comments as an attachment file.

Comments on the Quality of English Language

Largely okay but a once off proof-reading is recommended.

Author Response

The paper titled: "Output Feedback Integrated Guidance and Control Design for AutonomousUnderwater Vehicles Against Maneuvering Targets" was reviewed with mostly minor corrections bothering on the use of tenses. Some of these are marked as presented below.

Comments 1: [Abstract:

These limitations are further exacerbated in AUVs ("AUVs" Should be defined when used for the first time) due to unknown wave/current disturbances, harsh underwater acoustic conditions, and limited sensor capabilities. To address these challenges, this paper studies (studied) an integrated...". A model transformation is (was) introduced to synthesize...."Using these estimates from the observer, a finite-time event-triggered sliding mode controlleris (was) developed..."

1.Introduction

Line 67: The contributions of this paper are mainly (in) threefold:

Line 487: A finite-time EsO is (was) developed to...

Line 488: An event-triggered sliding mode controller is (was) then designed....

Kindly read through the document to track tenses related issues.

The "aneuvering targets" is a keyword in the research title and randomly mentioned from time to time. However, it is not clear how this phrase was catered for in the results (the graphs) presented.]

Response 1: [Thank you for the valuable comments. We have made more modifications to the tense, sentence structure, grammar and punctuation in the text. Thank you again for your careful review.]

Moreover, we have tried our best to check all the errors and made some changes in the manuscript. These changes will not influence the content and framework of the paper. And here we did not list the changes but marked in yellow in the revised paper. Thank you very much for the time and effort that you have spent in processing our manuscript. We wish the revision will meet the expectation from you, and addressed your concerns.

Thanks again for your kind help and encouragement.

Round 2

Reviewer 2 Report

Comments and Suggestions for Authors

I am satisfied with revisions